# Control cell migration by engineering integrin ligand assembly

Xunwu Hu [1,2], Sona Rani Roy[2], Chengzhi Jin[2,3], Guanying Li [2,4], Qizheng Zhang [2,5], Natsuko Asano[6], Shunsuke Asahina[6], Tomoko Kajiwara[7], Atsushi Takahara [7], Bolu Feng[8], Kazuhiro Aoki [9,10,11], Chenjie Xu[5] & Ye Zhang [1,2] ✉

Advances in mechanistic understanding of integrin-mediated adhesion highlight the importance of precise control of ligand presentation in directing cell migration. Top-down nanopatterning limited the spatial presentation to submicron placing restrictions on both fundamental study and biomedical applications. To break the constraint, here we propose a bottom-up nanofabrication strategy to enhance the spatial resolution to the molecular level using simple formulation that is applicable as treatment agent. Via self-assembly and co-assembly, precise control of ligand presentation is succeeded by varying the proportions of assembling ligand and nonfunctional peptide. Assembled nanofilaments fulfill multi-functions exerting enhancement to suppression effect on cell migration with tunable amplitudes. Self-assembled nanofilaments possessing by far the highest ligand density prevent integrin/ actin disassembly at cell rear, which expands the perspective of ligand-density-dependent-modulation, revealing valuable inputs to therapeutic innovations in tumor metastasis.

Cell migration plays a central role in a wide variety of biological phenomena, from embryogenesis to tumor metastasis, etc[1]. There is considerable interest in understanding cell migration on a molecular level because this could lead to novel therapeutic approaches in biotechnology[2]. Integrins, as the major family of cell receptors responsible for cell adhesion and migration, have long served as the primary targets of biomaterials[3–5]. Initially, the modulation of the adhesion surface relies on the control of the global density of integrin ligands[6]. Following the development of the polymer blending technique, the first generation of materials displaying multivalent ligands was synthesized, which signified the necessity of the regulation of ligand local density[7,8]. Two decades ago, fibronectin (50 μg/ml)-coated polystyrene microbeads were fabricated facilitating the elucidation of synergic effects of integrin occupancy and aggregation on cellular response[9,10]. About ten years ago, RGD-bound gold nanoparticles were fabricated as anchor points for single integrin $\alpha_v\beta_3$. Combined with block-copolymer micelle nanolithography, patterned surfaces with variable global ligand density ranging from 52 to 367 μm$^{-2}$

[1]Active Soft Matter Group, CAS Songshan Lake Materials Laboratory, Dongguan, China. [2]Bioinspired Soft Matter Unit, Okinawa Institute of Science and Technology Graduate University, Okinawa, Japan. [3]Guangzhou Municipal and Guangdong Provincial Key of Molecular Target & Clinical Pharmacology, the NMPA and State Key Laboratory of Respiratory Disease, School of Pharmaceutical Sciences and the Fifth Affiliated Hospital, Guangzhou Medical University, Guangzhou, China. [4]Institute of Medical Engineering, Department of Biophysics, School of Basic Medical Sciences, Health Science Center, Xi'an Jiaotong University, Xi'an, Shaanxi, China. [5]Department of Biomedical Engineering, City University of Hong Kong, Kowloon, Hong Kong SAR. [6]SM Application Group, JEOL Ltd., Tokyo, Japan. [7]Research Center for Negative Emission Technology, Kyushu University, Fukuoka, Japan. [8]Fluid Mechanics Unit, Okinawa Institute of Science and Technology Graduate University, Okinawa, Japan. [9]Division of Quantitative Biology, National Institute for Basic Biology, National Institute of Natural Sciences, Aichi, Japan. [10]Quantitative Biology Research Group, Exploratory Research Center on Life and Living Systems (ExCELLS), National Institutes of Natural Sciences, Aichi, Japan. [11]Department of Basic Biology, School of Science, SOKENDAI (The Graduate University for Advanced Studies), Aichi, Japan. ✉e-mail: zhangye@sslab.org.cn

and variable ligand spacing ranging from 50 to 135 nm were produced revealing the crucial influence of ligand spacing on $\alpha_v\beta_3$ integrin-mediated cell adhesion. Recently, via nanoimprint lithography, the RGD-bound 10-nm lines functioned as linear arrays of single integrin $\alpha_v\beta_3$ binding sites in single, crossing, or paired patterns with 40–490 nm distance were produced assisting the demonstration of ligand geometrical effects on adhesion cluster formation[11].

The development of nanofabrication techniques[12,13] enhanced control over the spatial presentation of integrin ligands, which promoted the mechanistic study of integrin-mediated adhesions to inspire biomaterial innovations. For instance, besides the global ligand density, the local ligand density, the size of 'ligand island', the spacing between islands, and the ligand spacing all became critical parameters in materials design to control cell adhesion. Undoubtedly, enhancing the control of ligand presentation beyond the current resolution of top-down nanofabrication for insights into subsequent cellular response will reveal design principles for future generations of biomaterials. To address the challenges, we develop a bottom-up fabrication strategy reaching molecular level resolution[14] by combining molecular self-assembly and co-assembly[15] for extracellular constructs. Compared to the top-down techniques, the proposed bottom-up strategy does not require high-cost equipment nor rigorous fabrication condition. Essentially, the simple formula of molecular assembly, as a practical and readily applicable approach, eases the boundary between fundamental study and biomedical applications.

## Results

### Engineering peptide assembly to control cell migration

As shown in Fig. 1a, our strategy is to covalently connect non-functional assembling motif to ECM-derived integrin ligand synthesizing an assembling ligand. The assembling ligands self-assemble into nanofilaments displaying super high ligand density. By mixing the non-functional assembling motif with assembling ligand at different proportions, the co-assembled nanofilaments displaying precisely controlled ligand densities are produced. To put our design into practice, repeats of L-phenylalanine (F, FF, and FFF) facilitating intermolecular aromatic interactions[16,17], were coupled to the N-terminus of IKLLI derived from laminin α1 chain[18] generating candidate assembling ligands. Compared with IKLLI, all three assembling ligands exhibited enhanced circular dichroism (CD) signals in the far UV region under aqueous condition, which indicated the assembly processes led to the formation of supramolecular structures. Specifically, both FIKLLI and FFFIKLLI self-assembled into random coil structure, while self-assembly of FFIKLLI formed β-sheet structure (Supplementary Fig. 1a–c). Intriguingly, these assembling ligands all exhibited enhanced integrin-binding affinity in solution. Among them, FFFIKLLI had the highest binding affinity (Supplementary Fig. 1d, e), and

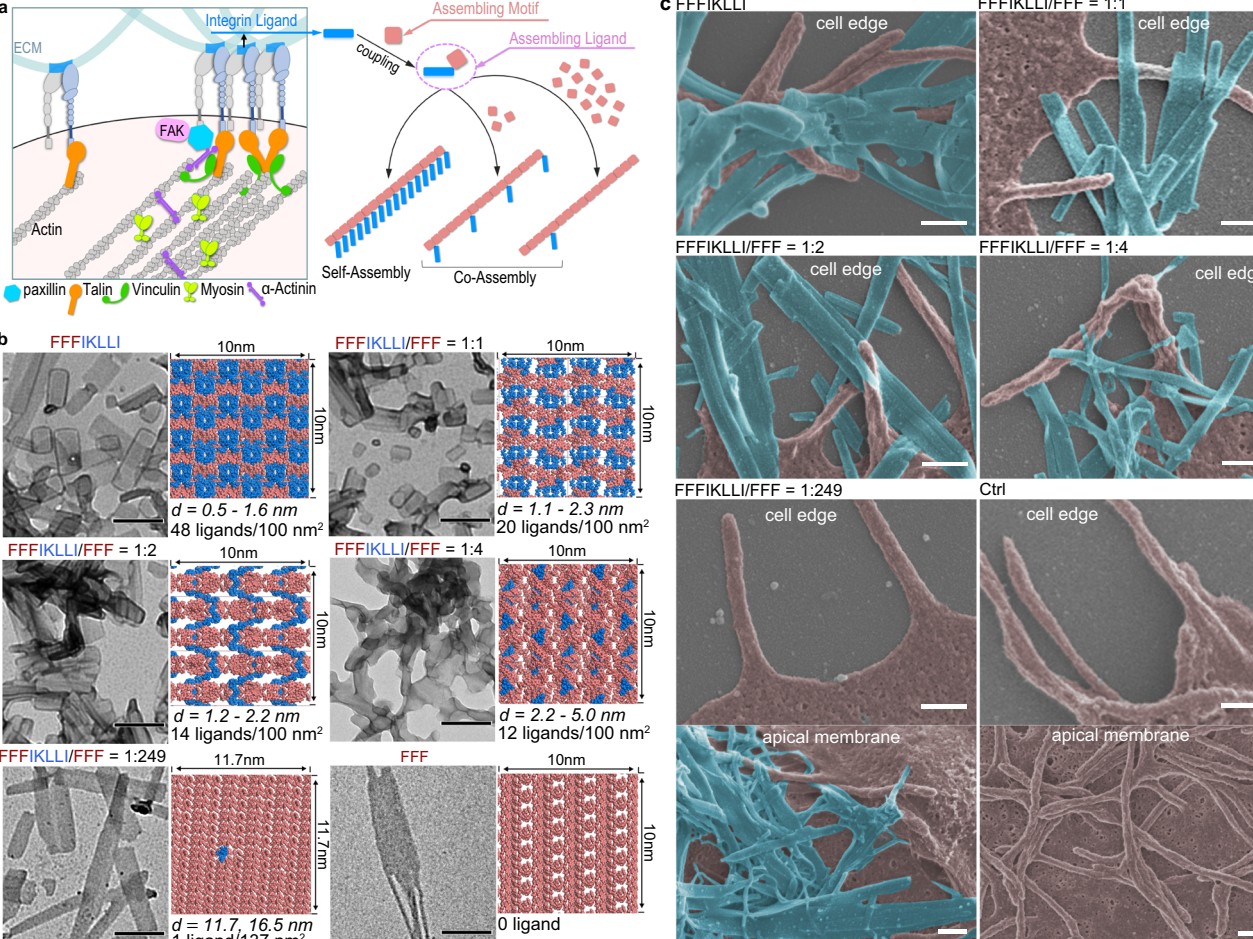

**Fig. 1 | Engineering integrin ligand assembly to control ligand presentation. a** Schematic illustration of precise control of integrin ligand presentation on nanofilaments via peptide assembly. **b** TEM images of nanofilaments obtained via molecular self-assembly and co-assembly of FFFIKLLI (100 μM) and FFF at various ratios, and the estimated molecular packing structures. IKLLI motif is presented in blue and FFF motif is presented in pink. The scale bars represent 200 nm. Three independent experiments were performed. **c** Zoom-in SEM images (false color) of HuH-7 cell edge and apical membrane after 3-day incubations. FFFIKLLI was maintained at a concentration of 100 μM. The cell body is highlighted in pink, while the nanofilaments are highlighted in blue. The scale bars represent 300 nm.

selectively targeted integrin $\alpha_3\beta_1$ (Supplementary Fig. 1f)[19], a molecular marker of malignant carcinomas. FFF, the non-functional assembling motif exhibited no obvious integrin $\alpha_3\beta_1$ binding affinity (Supplementary Fig. 1d).

To examine the integrin binding affinity in vitro, we tested FFFIKLLI on multiple malignant cancer cell lines expressing integrin $\alpha_3\beta_1$ (HuH-7, HeLa, HepG2, A549, MKN1, and U-87 MG), and cell lines lacking integrin $\alpha_3\beta_1$ expression (MCF-7 and Ect1/E6E7) using wound-healing assay and cell adhesion assay. By adding FFFIKLLI into adhered cell culture, without inducing cytotoxicity (Supplementary Fig. 2a), FFFIKLLI exerted a suppression effect on the migration of integrin $\alpha_3\beta_1$ expressing cells (Supplementary Figure 2b–g), but a negligible influence on MCF-7 and Ect1/E6E7 cells (Supplementary Fig. 3). After seeding cells on FFFIKLLI-coated dishes, cells expressing integrin $\alpha_3\beta_1$ remained adhered to the substrate while cells lacking integrin $\alpha_3\beta_1$ expression were washed out (Supplementary Fig. 4a–f). Knockdown of the $\alpha_3$ or $\beta_1$ integrin subunit of HuH-7 cells attenuated the suppression efficacy of FFFIKLLI (Supplementary Fig. 4g, k) and significantly reduced their adherence to the FFFIKLLI coated substrate (Supplementary Fig. 4l, m), while knockdown of the $\alpha_6$ subunit did not. The experimental results confirmed that FFFIKLLI selectively inhibits malignant cancer cell migration via targeting integrin $\alpha_3\beta_1$, not benign cancer cells lacking integrin $\alpha_3\beta_1$ expression.

Consistent to the MicroScale Thermophoresis characterization, without targeting integrin $\alpha_3\beta_1$, FFF didn't affect HuH-7 cell migration or adhesion (Supplementary Figure 2c, 5), which made it the qualified non-functional assembling motif for our proposed nanofabrication. By fixing the concentration of FFFIKLLI at 100 µM, we raised the proportion of FFF in a wide range to fabricate co-assembled nanofilaments (Supplementary Fig. 6) displaying different ligand densities for the preliminary evaluation. Self-assembly of FFFIKLLI and co-assembly of FFFIKLLI with FFF formed stable rectangular nanofilaments (~100 nm width, ~100–500 nm length) in water (Fig. 1b, Supplementary Figure 7). Compared with self-assembly, co-assembly gradually attenuated the suppression effect on cell migration by raising the proportion of FFF (Supplementary Figure 8). When the ratio of FFF to FFFIKLLI reached up to 44:1, co-assembled nanofilaments exhibited negligible influence on HuH-7 cells. Intriguingly, continued growth in the proportion of FFF gradually enhanced the cell migration. When the ratio of FFF to FFFIKLLI reached up to 249, more than 1.5 times faster cell migration was detected. No matter whether blocking or establishing the integrin-ECM adhesion, the regulation effect of nanofilaments was not affected (Supplementary Figure 9), which indicated that FFFIKLLI could fully regulate integrin $\alpha_3\beta_1$ without cell-ECM adhesions.

Quantitative estimation of surface ligand density of nanofilaments was conducted via molecular dynamics simulation based on the crystal structure of FFF unit cell[16], followed by polymorph prediction[20]. After the initial search, Fourier-transform infrared (FTIR) spectra of nanofilaments, which indicated the hydrogen bonding transition from N-H…N to N-H…O[21] due to the increasing proportion of FFF (Supplementary Fig. 10) were applied to select the adaptive packing modes (Supplementary Figs. 11–13, Supplementary Table 1). The polymorph predictions (Supplementary Fig. 14) suggested that the molecular packing of self-assembled FFFIKLLI could possibly expose 48 ligands per 100 nm², which is the highest record of ligand density, with the shortest distance between ligands ranging from 0.5 to 1.6 nm. Co-assembly of FFFIKLLI with FFF leads to decreased ligand density with increased ligand distance. For example, from 1 to 1, to 1 to 4, and to 1 to 249 ratio, the estimated ligand density decreased from 20 to 12 to 3 ligands per 100 nm² with minimum ligand distance increased from 1.1 to 2.2 to 4.9 nm, respectively (Fig. 1b, Supplementary Figs. 14, 15). Considering their influence on cell migration, we here categorize the ligand presentation on nanofilaments into four levels. Self-assembled FFFIKLLI possesses super high ligand density; co-assembled FFFIKLLI and FFF at 1 to 1 and 1 to 2 ratios possesses high ligand density; co-assembly at 1 to 4 ratio

possesses intermediate ligand density; and co-assembly at 1 to 249 ratio possesses low ligand density.

Although activated integrin $\beta_1$ co-localized with various FFFIKLLI-containing nanofilaments on integrin $\alpha_3\beta_1$ expressing cells (Supplementary Fig. 16), possessing different ligand density, nanofilaments demonstrated a variety of intimacy to the cell edge, especially the finger-like projections (Fig. 1c, Supplementary Fig. 17). The correlated influence on the phosphorylation of FAK and paxillin in different manners (Supplementary Fig. 18) suggested a series of regulations of integrin-mediated signals. Upon treatment of various peptide assemblies, the super-resolution SEM images of HuH-7 cells exhibited that nanofilaments with super high ligand density almost entangled with all peripheral projections. Meanwhile, a significant reduction of FAK Y397 phorsphorylation and paxillin Y118 phosphorylation was detected. By reducing the ligand density, only part of the cell edge had an intimate association with nanofilaments leaving more and more cell projections untouched. Under the same conditions, elevated levels of phosphorylated FAK and paxillin were obtained. By reaching low ligand density, the nanofilaments mainly attached to the apical membrane covering microvilli while the whole cell edge was untouched. And further enhancement on phosphorylation of FAK and paxillin was achieved.

Because cell migration is tightly associated with cell morphology, we next investigated the influence of nanofilaments on cell spreading and characterized the correlated cell motilities (Fig. 2). Without treatment, HuH-7 cells were round and less spread on glass (Ctrl). Upon the treatment of nanofilaments with super high ligand densities, restricted HuH-7 cells exhibited reduced spreading area with tentacle-like actin extensions in all directions. By reducing the ligand density from high to intermediate level, HuH-7 cells gradually resumed the smooth cell edge correlated to their partially restored motility (Fig. 2a, b). Upon the treatment of nanofilaments with low ligand density, cells exhibited broad, flat lamellipodia correlated to almost 2 times enhancement on travel speed and persistence (Fig. 2c, d).

## Peptide assemblies regulate membrane dynamics and FA organizations

Cell migration was defined as a four-step cycle: protrusion, adhesion, traction, and retraction, requiring spatiotemporal ordinations of the cytoskeleton and extracellular adhesion[22]. Therefore, to assess the impact of nanofilaments on migration-correlated membrane dynamics, we performed a morphodynamical analysis by mapping the protrusion and retraction along the entire periphery in response to various nanofilaments over time (Fig. 3a, Supplementary Fig. 19). The kymographs clearly indicated that nanofilaments with super high ligand density restricted the formation of protrusions. By reducing the ligand density to the intermediate level, the restriction effect was gradually attenuated. The quantitative analysis of protrusion and retraction velocity (Fig. 3b, Supplementary Fig. 20) showed that super high ligand density caused a global loss of the cluster periphery dynamics, while low ligand density promoted the formation of protrusion leading to enhanced cell migration.

We next studied integrin-mediated adhesion, which is a critical regulator of cell migration. Compared with control cells exhibiting peripheral FAs associated with peripheral actomyosin bundles, nanofilaments with super high ligand density caused vanishment of peripheral FAs but induced the formation of small dot-like FAs associated with tousled F-actin aggregates (Fig. 3c, Supplementary Figs. 21, and 22). Compared with control, treated HuH-7 cells exhibited more than 60% reduction on traction stresses (Fig. 3d, Supplementary Fig. 23), which was consistent to a dramatic decrease of pMLC expression (Supplementary Fig. 24). By reducing the ligand density, peripheral FAs associated with peripheral actomyosin bundles were formed again, together with the formation of larger streak-like FAs distributed on ventral cell surface connected with stress fibers. Particularly, at 1 to

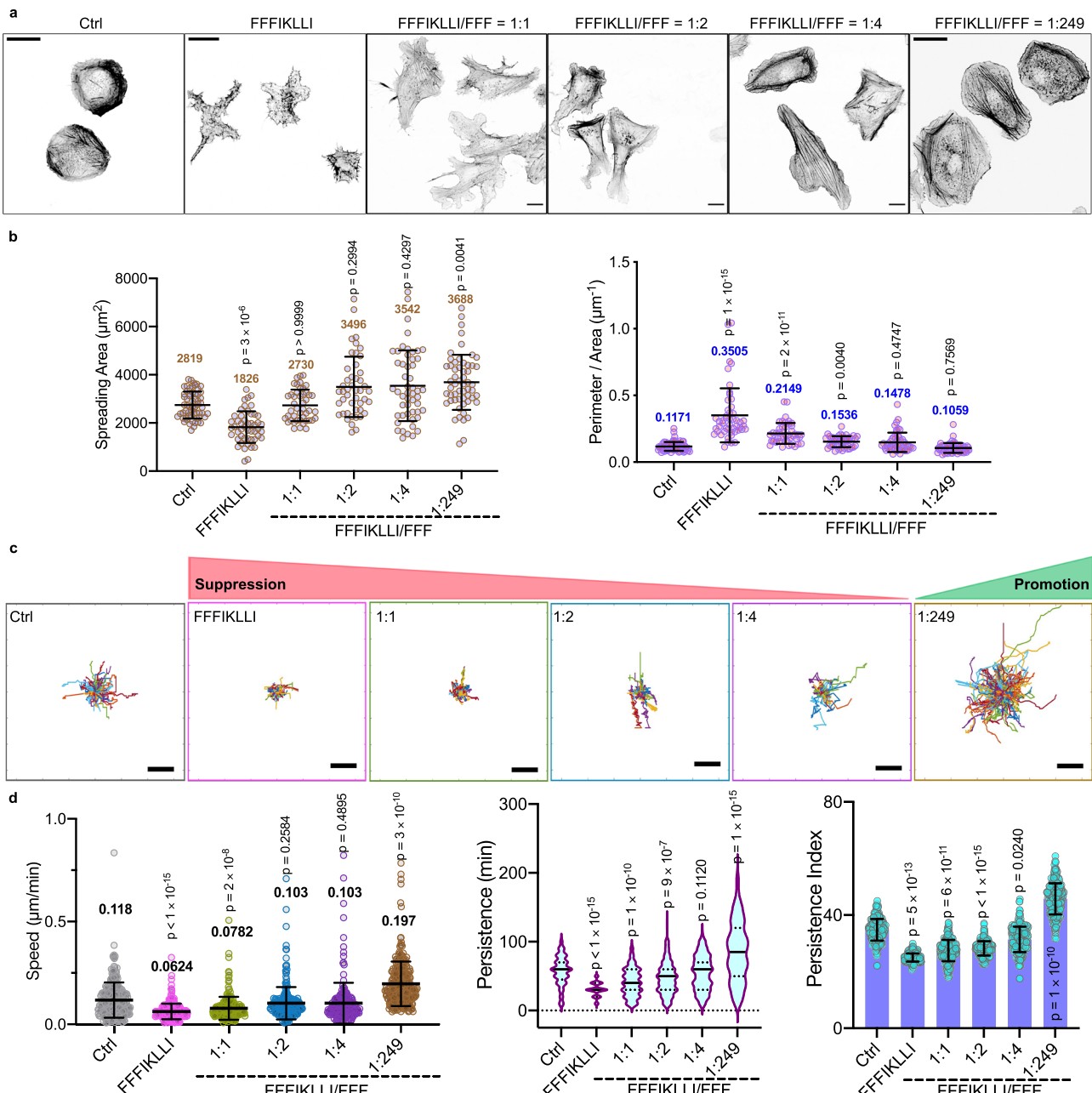

**Fig. 2 | Nanofilaments with various ligand presentations regulate both cell shape and cell migration. a** The phalloidin staining of Huh-7 cells with and without the treatment of various nanofilaments. Scale bars represent 20 μm. **b** The spreading area and the perimeter area ratio of HuH-7 cells under various conditions. Kruskal-Wallis with Dunn's multiple comparisons test was used for analysis of the data. Data are presented as mean ± s.d. From left to right, $n$ = 61, 49, 45, 46, 51, 56 cells, respectively. The trajectory plots **c**, and the correlated quantitative analysis of travel speed, persistence and persistence index **d** of randomly selected migrating cells for each incubation condition. Live cell images were taken every 10 min for a total of 10 h. Kruskal-Wallis with Dunn's multiple comparisons test was applied in data analysis. Error bars represent standard error of mean. From left to right, $n$ = 261, 280, 230, 260, 214, and 278 cells, respectively. Scale bars in panel **c** represent 50 μm. Source numerical data are available in source data.

2 ratio, the average traction stresses were resumed to the control level, while the maximal traction force exerted at the cell periphery with few stress foci localized to the protrusion area increased dramatically. At the intermediate level, the ligand presentation was optimal for the formation of ventral stress fibers inducing double enhancement of average traction stresses. The bipolar distribution of stress foci localized on the two lateral cell extremities indicated the altered actin organization polarized the cell to guide its migration[23]. With low ligand density, nanofilaments triggered the formation of nascent adhesions[24] localized across the lamellipodia and FAs increased in size toward the convergence zone associated with F-actin bundles collected in a

transverse band. Meanwhile, the traction force distribution exhibited a front-to-rear polarization with fused stress foci localized on lamellipodia corresponding to the distribution of FAs exerting 6 times higher retraction stresses compared with control, which indicated an enhanced cell migration.

## Nanofilaments with super high ligand density prevent trailing edge retraction

In response to nanofilaments with high, intermediate, and low ligand densities, HuH-7 cells exhibited three phenotype FA organizations corresponding to the cell migration velocities that have

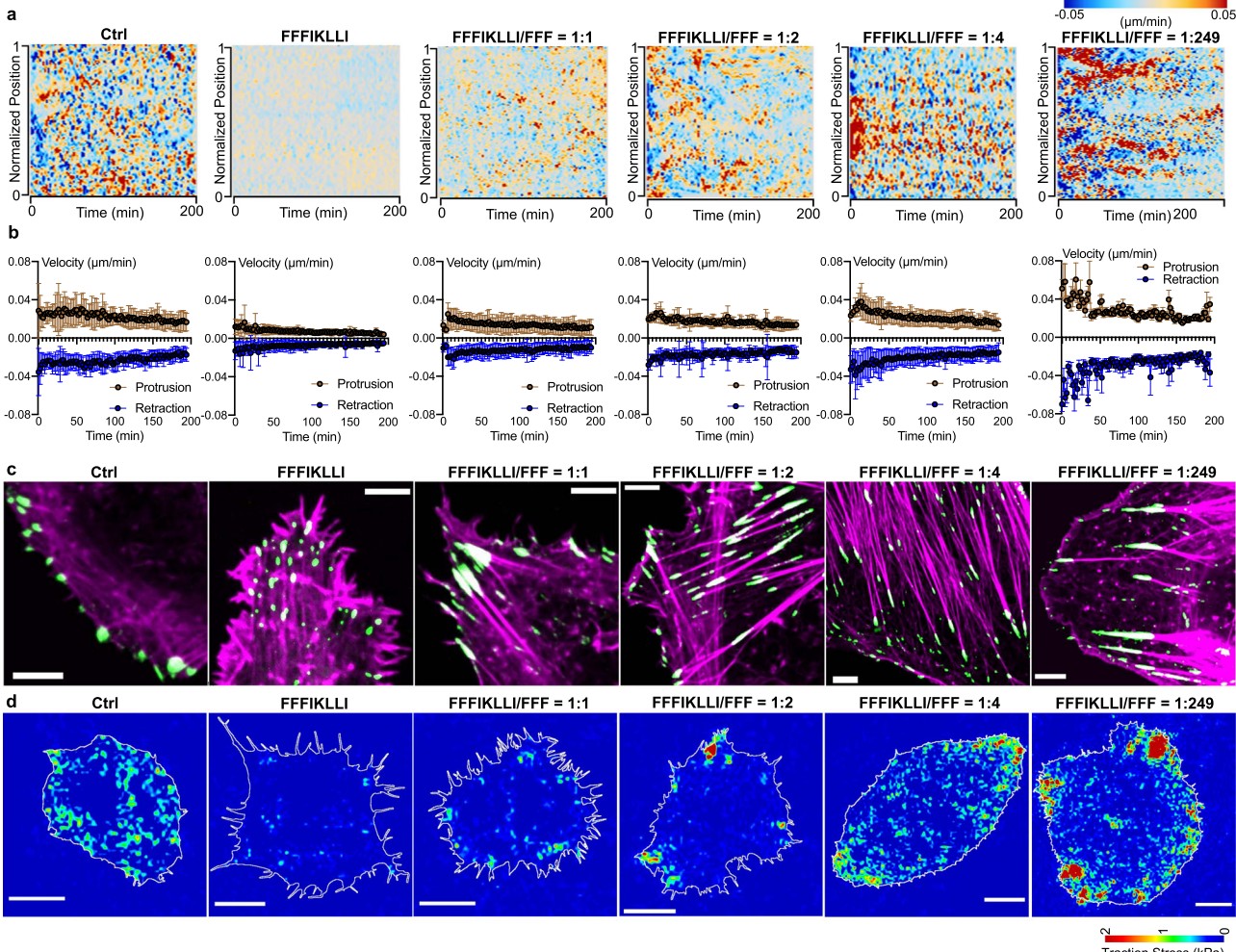

**Fig. 3 | Nanofilaments regulate membrane dynamics and FA organizations in a ligand density dependent manner.** Kymograph of normalized edge velocity **a** and mean velocity **b** over time for protrusions and retractions along the entire periphery of HuH-7 cells, $n = 8, 9, 9, 9, 10, 7$ cells, respectively. Data are presented as mean ± s.d. **c** Merged F-actin phalloidin staining, and paxillin immunofluorescence in HuH-7 cells. **d** Heat-scale plots of traction stress magnitudes of HuH-7 cells. At least three independent experiments were performed. Source numerical data are available in source data.

been well demonstrated on biomaterials with different ligand concentrations[25]. However, super high ligand density, which triggered an interesting interplay between F-actin and FAs that led to tentacle-like actin extensions on the cell periphery, has never been well-presented nor illustrated. By tracking actin and FA dynamics via time-lapse imaging of mRuby-Lifeact and FA components co-transfected HuH-7 cells (Supplementary Figs. 25–27), we observed that while the stress-fiber-associated FAs slide inward, the actin cytoskeleton at the cell rear was not fully disassembled (Fig. 4a). Different from the well-studied reduced trailing edge retraction caused by stable adhesion within the cell rear[26], we only observed the co-localization of integrin $\alpha_3\beta_1$ with talin and $\alpha$-actinin remaining on the actin filaments at the cell rear while vinculin, paxillin, and FAK located in the inward-sliding FAs connected with stress fibers (Fig. 4b, c, Supplementary Fig. 28, 29). Such segmentation of FA complex which led to failed FA disassembly on cell edge is highly possible due to the excessive binding interaction between integrin $\alpha_3\beta_1$ and ligands clustered in a super high density via self-assembly. Together with a great inhibition of Rac1 and RhoA activity on the cell periphery (Fig. 4d, e, Supplementary Fig. 30) which indicated the prevention of both protrusion formation and forward motion, it was demonstrated that self-assembly of FFFIKLLI restricted both trailing edge retraction and leading-edge protrusion of HuH-7 cells resulting into depolarization suppressing cell motility.

## Overcome the outside-in suppression of migration via inside-out activation

To understand the influence of super high ligand density on a molecular level, we did further exploration on the activation of intracellular signaling to overcome the outside-in restriction. In regard of the essential role of vinculin in the regulation of the assembly and disassembly of adhesion receptor complexes, we expressed vin258, a mutant that possesses vinculin D1 domain exhibiting high affinity to talin and paxillin but lack of actin-binding domain[27], in HuH-7 cells, to maintain united FA complex on the periphery even upon a 12 h treatment of FFFIKLLI (Fig. 5a, Supplementary Fig. 31a). However, solely allocating vinculin in FAs but without actomyosin-mediated forces directly acting on vinculin could not resume protrusion nor trailing edge retraction (Fig. 5b, Supplementary Fig. 31b–e). Eventually, the suppression effect on cell motility was remained (Fig. 5c, Supplementary Figure 31f–h). To preserve FAs on the cell edge with actomyosin-mediated force transmission, we applied Rho Activator II on HuH-7 cells to drive an elevated level of actomyosin contractility[28,29]. Followed by 12 hr treatment of nanofilaments, the FAs remained on the cell periphery associated with peripheral actin bundles (Fig. 5d). Enhanced contractile forces eased the full disassembly of FAs (Supplementary Fig. 32a, b) facilitating trailing edge retraction, which was indicated by the time-lapse images (Supplementary Figure 32a) and confirmed by the velocity profile of edge dynamics

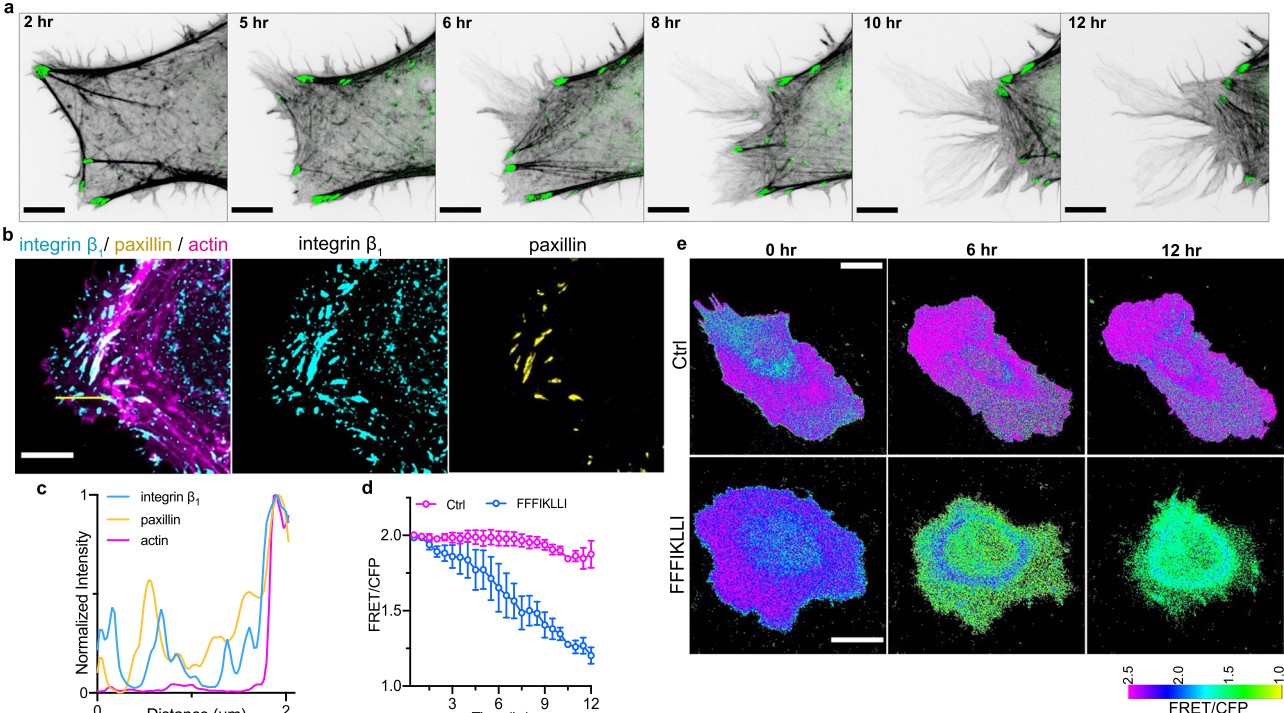

**Fig. 4 | Nanofilaments with super high ligand density induced uncoordinated FA dynamics preventing trailing edge retraction. a** Time-lapse series showing actin cytoskeleton (grey) and paxillin (green) in HuH-7 cells expressing mRuby-Lifeact-7 and mGFP-paxillin[49] upon the treatment of FFFIKLLI (100 μM) for 12 hr. Scale bars represent 2 μm. Three independent experiments were performed. **b** F-actin phalloidin staining (magenta), integrin β1 (cyan) and paxillin (yellow) immunofluorescence in HuH-7 cell after 12 h treatment of FFFIKLLI (100 μM). Scale bar represents 2 μm. **c** Fluorescence intensity distribution profile of integrin β1, paxillin, and actin cytoskeleton along the yellow line on merged image of **b**. **d** 12 h

time course Rac1 activity of HuH-7 cells with or without the treatment of FFFIKLLI (100 μM). Rac1 activity was measured by FRET. *n* = 6 cells (Ctrl) or 5 cells (FFFIKLLI). Symbols represent the mean FRET/CFP emission ratio ± s.d. **e** Representative FRET/CFP ratio images of HuH-7 cells expressing RaichuEV-Rac1 with or without the treatment of FFFIKLLI (100 μM) at the indicated time points and coded according to a pseudo-color scale, which ranges from yellow to purple with an increase in Rac1 activity. Scale bars represent 20 μm. Source numerical data are available in source data.

(Fig. 5e). However, the protrusion activity was still restricted resulting into partial restoration of cell motility (Fig. 5f, Supplementary Figure 32c-h). Because Rac1 can trigger new leading-edge formation by controlling local actin assembly and FA formation when activated at the cell edge[30], we then activated Rac1 constantly by translocating Tiam1 to the plasma membrane of HuH-7 cells to compel the formation of lamellipodia (Supplementary Figure 33a). Upon the treatment of nanofilaments for 12 hr, cells exhibited both FAs localized across the lamellipodia, and FAs associated with F-actin bundles collected in a transverse band (Fig. 5g). Both leading-edge protrusion and trailing edge retraction were not restricted (Fig. 5h, Supplementary Figure 33b, 34a-c), same as the cell motility (Fig. 5i, Supplementary Figure 34d-g). Together, the excessive binding interactions between integrins and the super high-density ligands globally deactivated the endogenous Rac1 via an outside-in path[31]. Besides removing the nanofilaments or lowering the ligand density on nanofilaments extracellularly, constant activation of Rac1/Tiam1 signaling was an effective intracellular rescue. Regarding Rac1's particular roles for tumor metastasis, the therapeutic potentials of super high ligand density are promising.

## Discussion

The conceptual exploration of constructing synthetic soft matters displaying highly ordered patterns with various symmetries is essential to the hierarchical design of advanced materials. In this research, we attempt to survey engineering efforts in molecular assembly, using synthetic chemistry-the rich tool set to create assembling building blocks, for the construction of bioactive materials via a bottom-up approach. Unlike most other self-assembled soft matters that utilize a single component, we introduced co-assembly to refashion the

molecular self-assembly into the advanced nanofabrication technique for the construction of structurally and functionally more complex materials achieving biphasic control of cancer cell motility (Fig. 5j). The association of functional building block in a precise manner pushes the boundary of nanofabrication producing super high ligand density via self-assembly surpassing the existing record inducing interesting cellular response that has never been reported. Via the exploration of fundamental biological questions, a novel therapeutic strategy against metastasis is unveiled.

To prove the concept, we randomly selected peptide sequences derived from ECM components possessing different binding interests to integrin isoforms for extracellular constructs[32,33]. Fibronectin-derived GRGDSP, LRGDN, and synergy peptide PHSRN, laminin β1 chain-derived YIGSR[34], and α1 chain-derived IKVAV, targeting integrin $α_5β_1$, $α_vβ_3$, $α_3β_1$, or $α_6β_1$, were coupled with FFF obtaining a series of assembling ligands for the construction of super high ligand densities[18]. Upon these treatments, cells phenocopied the morphology and motility of FFFIKLLI-treated cells (Supplementary Fig. 35) indicating that the design strategy could be applied as a generalized tool to fabricate biomaterials with therapeutic potentials in targeting subtype malignant tumors associated with different integrin expression pattern. Following the experimental observations, the insertion of mechanistic insights of dynamic interactions between the peptide assemblies and the integrin clustering from molecular dynamic simulations will guide the optimization of ligand engineering. Eventually, the advancement of molecular assembly may become a turning point of programmed matter to explore chemical modalities and to incorporate exciting findings in pharmacological biology.

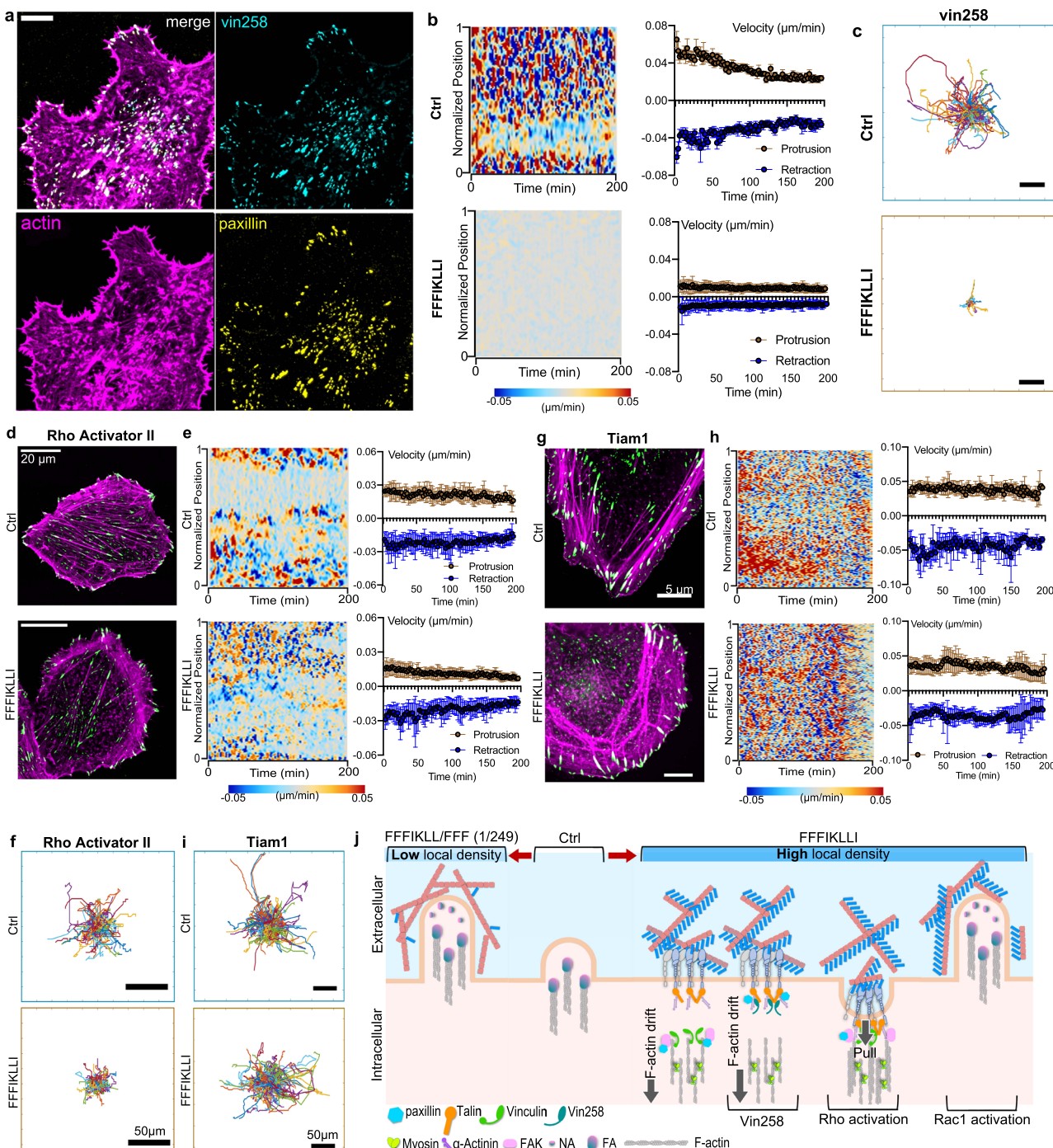

**Fig. 5 | Overcome the influence of super high density of ligands via Rac1 signaling. a** F-actin phalloidin (magenta), vinculin N-terminal domain lacking the tail domain (vin258) (cyan), and paxillin (yellow) immunofluorescence in HuH-7 cell expressing pEFGPC1/GgVcL 1-258 upon the treatment of FFFIKLLI (100 μM) for 12 h. Scale bar represents 5 μm. **b** Kymograph of normalized edge velocity and mean velocity over time for protrusions and retractions of HuH-7 cells expressing pEGFPC1/GgVcL 1-258 with or without the treatment of FFFIKLLI for 12 h. n = 10 cells. Data are presented as mean ± s.d. **c** The trajectory plots of ~200 randomly selected migrating HuH-7 cells expressing pEGFPC1/GgVcL 1-258 with or without the treatment of FFFIKLLI. Scale bars represent 50 μm. F-actin phalloidin staining (magenta) and paxillin (green) immunofluorescence (three independent experiments were performed) **d**, kymograph of normalized edge velocity and mean velocity over time for protrusions and retractions (n = 9, 12 cells, respectively, data are presented as mean ± s.d.) **e**, and the trajectory plots (n = ~200) **f** of HuH-7 cells pretreated with Rho Activator II (1 μg/mL) with or without 12 h treatment of FFFIKLLI (100 μM). F-actin phalloidin staining (magenta) and paxillin (green) immunofluorescence (three independent experiments were performed) **g**, kymograph of normalized edge velocity and mean velocity over time for protrusions and retractions (n = 7, 8 cells, respectively, data are presented as mean ± s.d.) **h**, and the trajectory plots (n = ~200) **i** of HuH-7 cells with elevated Rac1 activation with or without 12 h treatment of FFFIKLLI (100 μM). **j** Schematic summary of outside-in regulation of HuH-7 cell motility via precise control of ligand density on nanofilaments, and the restoration of cell motility via intracellular signaling. Source numerical data are available in source data.

## Methods

### Preparation of peptide assemblies

To prepare the stocks of peptide assembled nanofilaments, the required amount of peptide was dissolved in Milli-Q water at 100 mM (for FFF) or 10 mM (for the other peptides, and peptides mixture) and adjust the solution to reach a final pH of 7.0 using 1 N NaOH solution. To stabilize the peptide assemblies, the stock solutions were stored at room temperature for at least 24 h before application. The obtained nanofilaments were very stable and were not affected by simple dilution at neutral pH. Therefore, to prepare the working solution, the desired amount of stock solution was diluted using Milli-Q water or the culture medium. Stand the working solutions for 30 min to evenly distribute nanofilaments in the working solution before applying them to the experiments.

### Circular dichroism (CD) spectroscopy

CD spectra of the peptide assemblies in aqueous solutions were recorded under the nitrogen atmosphere using a spectrophotometer JASCO J-820 (Jasco, Japan). A quartz cuvette with 0.5 mm path length was used as a sample container. Continuous scanning mode is applied. And each sample is scanned three times to obtain an averaged spectrum.

### Fourier-transform infrared (FTIR) microscopy

To prepare FTIR samples, an aliquot of the peptide solution was dropped on a TS CaF2 window (5 mm diameter) (Edmund Optics, Japan) and freeze-dried using a lyophilizer (Freeze Dryer, Labconco). The CaF2 Window with assembled nanofilaments was mounted for FTIR microscopic analysis. FTIR microscopic analysis of a plain CaF2 Window served as the reference. An FTIR microscope (Nicolet iN10, Thermo Fisher Scientific) equipped with an MCT detector was used for collecting IR spectra. Spectra ranging from 4000 to 650 cm$^{-1}$ were obtained in transmission mode, aperture of 50 μm, with the resolution of 2 cm$^{-1}$. Each spectrum was derived from an average of 128 scans for each sample.

### Transmission electron microscopy (TEM) imaging of peptide assemblies

To prepare TEM samples, 5 μL peptide solution was dropped on a glow-charged copper grid (400 mesh) coated with carbon film. After removing excess solution, the grid was gently washed three times to remove excess nanofilaments. Then the sample grid was negatively stained using 1.0% (w/v) uranyl acetate and air-dried. TEM images were captured at a high vacuum using a transmission electron microscope (JEM-1230R, JEOL, Japan).

### Molecular dynamics simulation and polymorph prediction

Molecular mechanics calculations were performed using Materials Studio® 2020. For all simulations, the Ewald method[35,36] was used for the electrostatic and van der Waals interaction terms. Gasteiger charges were used for an initial conformational search. As the crystal structure prediction method uses a rigid body approximation in the initial search for crystal packing alternatives, the analysis to determine low energy geometry was performed by following the protocols reported by Kim, etc[37], and the results were used as input for the packing calculations. The conformation of FFF was reported by Ellenbogen, etc[16]. As reported, the single crystal structure of FFF in FFF-tape (self-assembled nanostructure) was determined by single crystal XRD measurements to 1.1 Å resolution. The determined structure is triclinic, space group P1, with four FFF molecules per asymmetric unit. The alignment of the unit cell with regard to the self-assembled tape structure reveals that growth is governed by π-interactions between adjacent aromatic rings along the c-axis. Based on the crystal structure of FFF, following a six-step process[38] as detailed in the Supplementary Information (Supplementary Figs. 11–14, Supplementary Table 1), the ligand density of peptide assembly was estimated.

### MicroScale thermophoresis

The interaction between integrins and peptide ligands in PBS was quantified by a Monolith® NT.115Pico (NanoTemper Technology, Germany) using MO. Control 1.6 software. Recombinant human integrin proteins, including alpha-3/ beta-1, alpha-5/beta-1, and alpha-6/ beta-1, were purchased from R&D Systems (USA). Recombinant integrin proteins were labeled by Monolith™ Protein Labeling Kit RED-NHS (NanoTemper Technology) following the instruction provided by the manufacturer. The fluorescence-labeled integrin solution was mixed with peptides solution in 15 concentration gradients at a 1:1 ratio. The Monolith™ NT.115 MST Premium Coated Capillaries (NanoTemper Technology) were applied to draw the mixed solution with the order of declining ligand concentration. Pretests were performed to ensure the protein concentration and fluorescent intensities in each tube were equal. Following that, quantitative binding affinity was measured three times.

### Cell culture and sample preparation

Human hepatocellular carcinoma cell lines HuH-7 (#RCB1366) and Hep G2 (#RCB1886), human gastric adenocarcinoma cell line MKN1(#RCB1003), human breast cancer cell line MCF-7 (#RCB1904) and human cervical cancer cell line HeLa (#RCB0007) were purchased from Riken BioResource Research Center.

Human lung carcinoma cell line A549 (#CCL-185), human glioblastoma astrocytoma cell line U-87 MG (#HTB-14), and human ectocervical cell line Ect1/E6E7 (#CRL-2614) were purchased from American Type Culture Collection (ATCC).

Huh-7 and Hela were cultured in Dulbecco's Modified Eagle Medium (DMEM, Gibco) supplemented with 10% fetal bovine serum (FBS, Gibco), penicillin (100 μg/mL), and streptomycin (100 μg/mL). MCF-7 was cultured in MEM (Gibco) with 10% FBS, penicillin (100 μg/mL), and streptomycin (100 μg/mL). Hep G2 was cultured in Minimum Essential Media (MEM, Gibco) containing 10% FBS, Non-Essential Amino Acids (0.1 mm, Gibco), penicillin (100 μg/mL), and streptomycin (100 μg/mL). U-87 MG was cultured in Eagle's Minimum Essential Medium (EMEM, ATCC) containing 10% FBS, penicillin (100 μg/mL), and streptomycin (100 μg/mL). MKN1 was cultured in RPMI 1640 medium (Gibco) containing 10% FBS, penicillin (100 μg/mL), and streptomycin (100 μg/mL). A549 cell line was maintained in Ham's F-12K (Kaighn's) medium (Gibco) with 10% FBS, penicillin (100 μg/mL), and streptomycin (100 μg/mL). Ect1/E6E7 was cultured in Keratinocyte-Serum Free medium (Gibco) with 0.1 ng/ml human recombinant EGF, 0.05 mg/ml bovine pituitary extract, and additional calcium chloride 44.1 mg/L (final concentration 0.4 mM).

For peptide treatment, stock solutions of peptide or the equivalent volume of DPBS were diluted in the culture medium containing low FBS (0.5%) and aged for 30 min before applying to the cells.

Congo red (Abcam) was freshly prepared in the culture medium containing 1% FBS at the concentration of 0.1 mg/ml. After aspirating the old culture medium, the freshly prepared Congo red solution was added to the cells and incubated for 30 min at 37 °C. The cells were washed with PBS for 3 times. The cells were then fixed with 4% formaldehyde for 10 min and washed 3 times with PBS. The fixed cells were then blocked with 5% BSA for 1 hr at room temperature before co-stained with phalloidin and antibodies.

For Rho activation, cells were treated with 1 μg/ml Rho Activator II (#CN03-A, Cytoskeleton, Inc.) for 4 h before applying to additional manipulations.

For transfection, cells were transfected by electroporation with 2.5-5 μg of endotoxin-free plasmid DNA per -1× 106 cells using the 4D-Nucleofector (Lonza) according to the manufacturer's instruction, except where indicated otherwise.

For the Rac1 activation, HuH-7 cells were transfected with Lyn11-linker-FRB, YFP-FKBP-linker-Tiam and mRuby-Lifeact-7 in a 2:1:1 ratio. One or two days after transfection, the cells are set to a time-lapse microscope, and treated with 100 nM Rapamycin. And the Tiam1 was translocated to the membrane within 2 mins to locally activated Rac1 signaling.

## Scanning electron microscopy (SEM) imaging of peptide assemblies treated HuH-7 cells

For the preparation of the SEM samples, HuH-7 cells in the exponential growth phase were seeded on a 35 mm glass bottom petri dish. Once the cells were fully attached, peptide assemblies were added to the cell culture. After 12 h incubation, the culture medium was removed, and the cell culture was washed three times using 1×PBS buffer. 2.5% glutaric dialdehyde was added to the cells for 30 min followed by 1% osmium to fix the cells. Then the cells were dehydrated using ethanol, rinsed with t-BuOH, and freeze-dried in a lyophilizer (Freeze Dryer, Labconco) for more than 12 h. All samples were coated with 5 nm Osmium (Os) using Os coating device OPC80T (Filgen) before imaging. And the SEM images were captured using an ultra-high-resolution FE-SEM JSM-IT800SHL (JEOL, Japan) at 1.0 kV WD 6 mm.

## 3-(4,5)-dimethylthiahiazo (-z-y1)−3, 5- diphenylte- trazolium bromide (MTT) assay

Huh-7 cells were seeded in 96-well plates at a density of $5 \times 10^3$ cells/well. The cells were allowed to adhere by incubating at 37 °C with 5% $CO_2$ for 12 hr. 100 μL of Cultured medium containing 10% FBS and the desired concentration of peptide replaced to each well and the cells were then incubated with the peptide solution for another 24, 48, or 72 hr. After the desired time of incubation, 10 μL of MTT (12 mM, Invitrogen) solution at a concentration of 5 mg/mL was added to each well and incubated at 37 °C for another 4 hr. The reduction reaction was terminated by adding 100 μL of 10% of SDS solution (in Milli-Q) and incubated for 12 hr to dissolve the formazan crystals formed in the cells. A Nivo multimode plate reader (PerkinElmer) was used to measure the optical density at the 570 nm wavelength of each well. All experiments were conducted in triplicate, and the results were calculated as mean ± standard deviation and presented as cell viability.

## Wound healing assay

Healthy cells were seeded in 96-well Image Lock plates (Essen Bioscience, UK). Each well contained $2 \times 10^4$ to $5 \times 10^4$ cells in 100 μL of complete cell culture medium. Cell monolayer with approximately 90% confluence formed after 12 hr and cells were next starved in serum-free culture medium for 12 hrs. Homogenous scratch wounds with a width of about 700 μm were made by an Essen BioScience-Wound Maker. The detached cells were washed with Dulbecco's Buffered Saline (DPBS, Gibco) 3 times. Cells were incubated with 100 μL of culture medium containing 0.5 % FBS and desired concentration of each peptide. Wound closure was monitored by an IncucyteS3 (Essen Bioscience) with a 10x objective. Images for each well were acquired every 2 hr to 8 hr for 2 days or until the wounds were closed completely. The wound healing rate was quantified by Incucyte Scratch Wound Analysis Module (Essen Bioscience). The experiments were repeated at least 3 times, and the results were presented as mean ± standard deviation of at least 3 independent experiments.

## Western blotting

Protein lysates were prepared in ice-cold RIPA lysis, and extraction buffer (Thermo Scientific) supplemented with 1% Halt Protease Inhibitor (Thermo Scientific) and 1% Halt Phosphatase inhibitor (Thermo Scientific). Cells were scraped off from the culture surface using a cold plastic cell scraper, and the cell suspension was then gently transferred into a pre-cooled microcentrifuge tube and maintained on ice for 30 min with constant agitation before centrifuged at $21000 \times g$ for

20 min. The supernatant was then transferred to a new tube, and the protein concentration of the supernatant was quantified according to the Pierce™ BCA Protein Assay Kit (Thermo Scientific). The rest of the supernatant was mixed with 4x laemmli sample buffer (Bio-Rad) in a 1:3 ratio and boiled at 100 °C for 10 min. Lysates can be aliquoted and stored at −80 °C freezer. When running the gel, 30 μg of protein was loaded into each well of the SDS-PAGE gels, along with a molecular weight marker (Bio-Rad). The gel was run for 2 hr at constantly 100 V to separate the proteins based on the molecular weight. The proteins were then transferred from the SDS-gel to a PVDF membrane (Bio-Rad) using a Transfer-blot turbo system (Bio-Rad). Once the membranes were blocked with Blocking One P solution (Nacalai) for 20 min at room temperature with gentle shaking, they were probed with antibodies against GAPDH (6C5, ab8245, Abcam, 1:1000), integrin beta 1 (12G10, ab30394, Abcam, 1:100), integrin alpha 3 (ASC-1, MA5-28565, Invitrogen, 1:100), integrin alpha 6 (EPR18124, ab191551, Abcam, 1:200), Phospho-Myosin Light Chain 2 (Thr18/Ser19)(pMLC, #3647, Cell Signaling Technology, 1:200) diluted in Tris-Buffered Saline containing 0.1% Tween-20 and 5% Blocking one P solution and the membrane was incubated in the dark for overnight at 4 °C with gentle shaking. The unbounded antibodies were washed out with tris-buffered saline (TBS) containing 0.1% Tween-20 (TBS- T) for 10 min for at least 6 times. The membranes were further incubated with secondary antibody [conjugated with horseradish peroxidase (goat anti-mouse: #G-21040, goat anti-rabbit: #31460, Invitrogen) for 1 hr at room temperature. The membranes were rewashed with TBS-T for 10 min for at least 6 times. The protein bands were visualized according to the Chemiluminescence ECL detection kit (Bio-Rad) with a Fujifilm/GE LAS-3000. Protein bands were quantified by measuring peak areas using ImageJ. The peak area for each protein was normalized against the peak area of the loading control.

## Quantitative RT-PCR

Total RNA was immediately extracted from cultured cells using TRIzol reagent (Invitrogen). cDNA was synthesized with the cDNA Synthesis Kit (Bio-Rad). qRT-PCR was performed on a qTOWER 3 Real-Time Thermal Cyclers (Analytik Jena) with qPCRsoft 3.2 software using Power SYBR Green Master Mix (ThermoFisher Scientific). Gene expression of integrin was normalized using the comparative Ct quantification method. And the primer sequences were used as follows:

hGAPDH: 5'-GGCATCCTGGGCTACACTGA-3'(F), 5'-GAGTGGGTGT CGCTGTTGAA-3'(R); hITGA3:5'-GCCTGACAACAAGTGTGAGAGC-3' (F), 5'-GGTGTTCGTCACGTTGA TGCTC-3' (R); hITGB1: 5'-GGATTCTCCA-GAAGGTGGTTTCG-3' (F), 5'-TGCCACCAAGT TCCCATCTCC-3' (R).

## Integrin knockdown constructs and fluorescent protein fusion constructs

Integrin β1 shRNA plasmid (#sc-29375-SH), Integrin α3 shRNA plasmid (#sc-35684-SH), Integrin α6 shRNA plasmid (#sc-43129-SH), and control shRNA plasmid-A (#sc-108060) were purchased from Santa Cruz Biotechnology for knockdown of the target integrins. The knockdown efficiency was evaluated by western blotting.

The mGFP-paxillin expression vector was a gift from A. Kusumi (Okinawa Institute of Science and Technology, Japan). pGFP-FAK and pEGFP-vinculin were gifts from Kenneth Yamada (RRID: Addgene_50513 and Addgene¬_50515)[39], EGFP-talin, mVenus-Integrin-Beta1 and mRuby-Lifeact-7 was a gift from Michael Davidson (RRID: Addgene_54560). pEGFPC1/Gg Vinculin 1-258 (akaVD1) was a gift from S. Craig (RRID: Addgene_46270)[40]. The Rac1 FRET Sensor, pCAGGS-RaichuEV-Rac1was generously provided by K. Aoki (National Institute for Basic Biology, Japan)[41] and the RhoA FRET sensor, DORA-RhoA was generously provided by Yi I. Wu (University of Connecticut Health Center, USA)[42]. The expression vectors for Rac1/Tiam1 activation system (Lyn11-linker-FRB, YFP-FKBP, YFP-FKBP-linker-Tiam1, and pTriEx-

PA-Rac1) was generously provided by T. Inoue (Johns Hopkins University, USA)[43].

## Flow cytometry

Cell monolayers were digested to obtain single-cell suspensions. Antibodies against integrin beta 1 (12G10, ab30394, Abcam, 1:200), integrin alpha 3 (ASC-1, #MA5-28565, Invitrogen, 1:100) or Isotype controls (mouse IgG, ab37355, Abcam, 1:100) was added to the single cell suspensions and incubated the cell suspensions on ice for 30 min. Cell suspensions were then washed with cold DPBS twice for 10 min before incubated with the Alexa Fluor488-conjugated secondary antibody (mouse IgG, ab150113, Abcam, 1:1000) for another 30 min on ice. After washed with DPBS twice, the cell suspensions were loaded into Imaging Flow Cytometer (ImageStream X Mark II, Merck) to count the cell populations using Amins IDAS 6.0.

## Cell adhesion assay

The 96-well microplates wells were coated with peptide assemblies (the concentration of FFFIKLLI was kept at 100 μM constantly) for 12 hr at 37 °C before blocked with the Blocking Buffer (2% BSA, 1 mM CaCl2 and 1 mM MnCl2 in PBS) for 1 hr at 37 °C. Cells were collected from culture dishes and suspended in the Blocking Buffer containing anti-laminin-5 (P3H9-2, MAB1947, Chemicon, 5 μg/ml), anti-fibronectin (IST-9, ab6328, Abcam, 20 μg/ml) or IgG isotype control (20 μg/ml, Abcam) before immediately seeded to the coated well (20,000 cells/well) and allowed to incubate at 37 °C for 1 h. The wells were washed for three times with the Blocking Buffer and the phase-contrast images were captured by IncucyteS3 (Essen Bioscience) with a 10× objective.

## Confocal Microscopy and image analysis

All microscope imaging was performed with a Zeiss LSM 780 or Olympus SD-OSR. To assess the F-actin organization after certain treatments, cells were seeded on a glass-bottom culture dish (D11130H, Matsunami). After culture with the peptides, cells were fixed with 4% paraformaldehyde phosphate buffer solution (PFA, 30525-89-4, Wako) for 10 min at room temperature, washed, and permeabilized with 0.1% Triton X-100 (Sigma) in PBS for 15 min. Cells were washed with DPBS twice and incubated with ActinRed (Rhodamine-conjugated phalloidin, R37112, Invitrogen) or ActinGreen (Alexa-Fluor-488-conjugated phalloidin, R37110, Invitrogen) for 15 min at room temperature. Cells were washed with DPBS before being imaged. Morphological characteristics, including cell spreading area and perimeter area ratio, were quantified by Image J.

For live-cell time-lapse imaging, transfected HuH-7 cells were placed inside a stage-top incubator (Tokai Hit) and were tracked with a 100x/1.35 Silicon UPlanSApo objective for more than 12 h. 5.5-μm stack images were acquired every 20 min, and imaging parameters were adjusted to minimize photobleaching and avoid cell death.

## Immunocytochemistry

After 4% PFA fixation, cells were blocked by 5% bovine serum albumin (BSA, A7906, Sigma) in DPBS containing 0.1% Triton X-100 for 1 h, followed by the incubation of primary antibodies against integrin beta 1 (12G10, ab30349, Abcam, 1:200), paxillin (Y113, ab32048, Abcam, 1:200), CD49c (integrin alpha 3, ASC-1, MA5-28565, Invitrogen, 1:50), talin 1 (8D4, ab157808, Abcam, 1:100), vinculin (EPR8185, ab129002, Abcam, 1:100), FAK (#3285, Cell signaling Technology, 1:200), α-actinin (H-2, sc-17829, Santa Cruz Biotechnology, 1:200), and Phospho-Myosin Light Chain 2 (Thr18/Ser19) (pMLC, #3674, Cell Signaling Technology, 1:100) in 1% BSA for overnight at 4 °C. The samples were washed twice with PBS before applying fluorescence-conjugated secondary antibodies, including Goat Anti-Mouse lgG H&L (Alexa Fluor® 488) (ab150113, Abcam, 1:1000), Goat Anti-Rabbit lgG H&L (Alexa Fluor® 488) (ab150077, Abcam, 1:1000), Goat Anti-Rabbit lgG H&L (Alexa Fluor® 568) (ab175471, Abcam, 1:1000), Donkey Anti-Rabbit lgG H&L (Alexa Fluor® 647) (ab150075, Abcam, 1:1000), Goat Anti-Mouse lgG H&L (Alexa Fluor® 568) (ab175473, Abcam, 1:1000), Goat Anti-Mouse lgG H&L (Alexa Fluor® 647) (ab150115, Abcam, 1:1000), in 1% BSA with or without dye-conjugated phalloidin and DAPI (R37606, Invitrogen) at room temperature for 45 min.

## Random cell migration assay

HuH-7 cells were labeled with fluorescent protein fusion constructs or Hoechst 33342. After being treated with or without the peptides for 12 hr, the cells were tracked with a 20x/0.8 Plan-Apochromat objective for 6 h. The centroids of labeled cells were tracked using the Track-Mate plugin (https://imagej.net/plugins/trackmate/) in ImageJ[44]. A custom MATLAB script generated by Mr. B. Feng was applied to reconstruct the cell migration trajectory. For tracking the cell migration, HuH-7 cells were labeled with fluorescent protein fusion constructs or Hoechst 33342. After being treated with or without the peptides for 12 h, the cells were tracked with a 20x/0.8 Plan-Apochromat objective for 6 h. The centroids of labeled cells were tracked using the TrackMate plugin (https://imagej.net/plugins/trackmate/) in ImageJ. A custom MATLAB script generated by Mr. B. Feng was applied to reconstruct the cell migration trajectory.

The code used for the Random Cell Migration Assay can be retrieved here: https://github.com/XunwuHu/source-code-for-nucleus-tracking.

The directionality of the cell movement was described by the persistence index and the persistent time, the persistence index was defined by the ratio of the vectorial distance (the distance between the origin and the endpoint of the movement) and the length of the total path and the persistent time is defined as the time it takes for the cell to change the initial direction by 90° [45].

## Morphodynamics analysis

The whole-cell morphodynamics maps were generated using the open-source ImageJ plugin "ADAPT"[46]. The analyses were done on the maximum intensity projection of 5.5-μm stack images of mRuby-Lifeact-7 transfected HuH-7 cells pretreated with peptides for 12 hr. A spinning-disk confocal (Olympus SD-OSR) equipped with the camera Prime BSI sCMOS camera and a 100×/1.35 Silicon UPlanSApo objective was used to obtain the images every 2 min. All velocity values are directly measured using the plugin and plotted using GraphPad Prism. Briefly, the fluorescent images were Gaussian filtered to suppress noise and then a gray-level threshold was applied to create a binary image. The cell boundary was taken as pixels bordering segmented regions and the resulting segmentation is used as the seed for the region-growing algorithm in the next frame. Velocity was calculated at each point on the cell boundary based on the change in gray level between two frames as the protrusion resulted in an increase in gray level and the retraction induced a decrease in gray level at a particular spatial coordinate over time. The change in gray level was used to calculate the membrane velocity at each point and for a better visual representation, the resulting velocity map images (around 10000 × 100 pixels) were stretched to optimize visualization. For a better visual representation, the map images (around 10000 × 100 pixels) were stretched to optimize visualization.

## Traction stress analysis

mRuby-Lifeact-7 transfected HuH-7 cells were applied for the traction force microscopy experiments. The in situ fluorescent images of the cell was applied to define the area of measurement. The cells were seeded on a PDMS substrate functionalized with 0.2 μm FluoSpheres™ Carboxylate-Modified Microspheres (F8807, Invitrogen) at a density of approximately 6000 cells/cm². The substrate was fabricated by following the published protocol on (STAR) Protocol[45], except that there were no ECM proteins coated on the substrate here. The stiffness of the PDMS substrate was characterized using a compression test, and

the results were summarized in Supplementary Figure 36, which indicated that the Young's modulus of the PDMA substrate was $12.1 \pm 0.29$ KPa. After being pretreated with peptides for 12 hr, the samples were imaged by a 100×/1.35 Silicon UPlanSApo objective, and the bead displacement obtained from confocal imaging was converted into force-displacement fields following established protocol[47,48]. Briefly, Images of beads with and without cell attachment were first aligned to correct experimental drift using ImageJ plugin "align slices in stack". The displacement field was subsequently calculated by another ImageJ plugin "PIV (Particle Image Velocimetry)". The cross-correlation PIV with a size of $64 \times 32$ pixels was used on all images for the PIV analysis to produce the position and vector field of the bead displacement. With the displacement field obtained from the PIV analysis, the traction force field was then reconstructed by the ImageJ plugin "FTTC". The correlated lifeact images were used to define the cell area and the traction boundary of the stress.

### Synthesis
All the solvents and chemicals are commercially available. Chemicals were used without further purification: solvents were further purified by Ultimate Solvent System (Nikko Hansen, Japan) before use. $^1$H NMR spectra were recorded in deuterated solvent on a Bruker Advance 500 MHz spectrometer, and a JEOL JNM-ECZR 600 MHz spectrometer.

Seven peptides (FFF, FFIKLLI, FFFIKLLI, FFFKLIIL, FFFGRGDSP, FFFLRGDN, FFFIKVAV) (Supplementary Figs. 37–49), plus all Fmoc-amino acids and resin that used in the peptide synthesis, were purchased from GL Biochem (Shanghai) Ltd. China. Peptides IKLLI, FIKLLI, FFFPHSRN and FFFYIGSR were synthesized on a peptide synthesizer (Intavis Bioanalytical Instruments) by following the Fmoc-based solid phase peptide synthesis principle: 2-chlorotritylchloride resin2 (0.63 g, 2 mmol) was swollen in two volumes of DCM for 30 min and washed with DMF for 3 times. Amino acids are Fmoc-Ile-OH (2.12 g, 6 mmol), Fmoc-Leu-OH (2.12 g, 6 mmol), Fmoc-Lys(Boc)-OH (2.81 g, 6 mmol), Fmoc-Gly-OH (1.78 g, 6 mmol), Fmoc-Arg(Pbf)-OH (3.89 g, 6 mmol), Fmoc-Pro-OH (2.02 g, 6 mmol), Fmoc-His(Trt)-OH (3.72 g, 6 mmol), Fmoc-Ser(tBu)-OH (2.30 g, 6 mmol), Fmoc-Asn(Trt)-OH (3.58 g, 6 mmol), and Fmoc-Phe-OH (2.32 g, 6 mmol) used for the synthesis were 3 times of resin in mmol. Each amino acid was added to DIPEA (1.55 g, 12 mmol) and dissolved in 30 mL of DMF. Based on the peptide sequences, the amino acid solution was added to the resin and the subsequent step. Each coupling step was run for 20 min and washed with DMF for 3 times. A mixture of DIPEA:MeOH:DCM (5:15:80) solution was added and reacted for 10 min. NMM (3.480 mL, base), NMP (0.232 μL, 24 mmol) and HBTU (2.27 g, 6 mmol) were used for activating the carboxylic group of the subsequent amino acid. The Fmoc was removed with a 20% piperidine solution in DMF. After the desired length of a peptide, the Fmoc and the side protected groups were cleaved with 95% TFA in water. The resulting peptides were precipitated 5 by cold ether. The precipitant was vacuum dried followed by lyophilized. The TFA in the dried powder was removed by dissolving in 0.1% acetic acid and freeze-dried. Acetic acid was removed with water and freeze-dried. The final products were obtained as white powders[18].

### Statistics and reproducibility
No statistical methods were used to predetermine sample size. The experiments were not randomized, and the investigators were not blinded to allocation during experiments and outcome assessment. All measurements were performed on 1–3 biological replicates from separate experiments. The exact sample size and exact statistical test performed for each experiment are indicated in the appropriate figure legends. Statistical analyses were performed using GraphPad Prism (GraphPad Software, www.graphpad.com). All bar graphs show mean values with error bars (s.e.m. or s.d., as defined in legends). The reported $P$ values were corrected for multiple comparisons, where appropriate. Precise $P$ values are shown in the figures and, when appropriate, are rounded to the nearest single significant digit. $P$ values less than 0.0001 maybe be provided as a range. $P$ values less than 0.05 are considered to be significant.

### Reporting summary
Further information on research design is available in the Nature Research Reporting Summary linked to this article.

## Data availability
All data generated in this study are provided in the Supplementary Information/Source Data file. Source data are provided with this paper.

## Code availability
The code used for the Random Cell Migration Assay can be retrieved here: https://github.com/XunwuHu/source-code-for-nucleus-tracking.

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

## Acknowledgements

The mGFP-paxillin expression vector was generously provided by A. Kusumi Lab at Okinawa Institute of Science and Technology, Japan. The Rac1 FRET Sensor, pCAGGS-RaichuEV-Rac1 was generously provided by K. Aoki Lab at NIBB, Japan. The expression vectors for Rac1/Tiam1 activation system (Lyn11-linker-FB, YFP-FKBP, YFP-FKBP-linker-Tiam1, and pTriEx-PA-Rac1) were generously provided by T. Inoue at Johns Hopkins University, USA. The RhoA FRET sensor, DORA-RhoA was generously provided by Yi I. Wu at University of Connecticut Health Center, USA. This work was supported by CAS Songshan Lake Materials Laboratory (Y.Z.), Takeda Science Foundation (2017 medical science) (Y.Z.), and JSPS Grant-in-Aid for Scientific Research (B) (21H02063, Y.Z.), and OIST Proof-of-Concept (POC) Program 2019 (Y.Z.).

## Author contributions

X.H. and Y.Z. conceived the study and designed the experiments. X.H., S.R.R., C.J., G.L., Q.Z., N.A., S.A., T.K., C.X., and Y.Z. performed the experiments. X.H. conducted circular dichroism spectroscopy, TEM imaging, microscale thermophoresis, MTT assay, wound healing assay, western blotting, quantitative RT-PCR, flow cytometry, cell adhesion assay, confocal microscopy, random cell migration assay, morphodynamic analysis, traction stress analysis; S.R.R., C.J., and G.L. conducted solid phase synthesis; Q.Z. and C.X. conducted the NMR spectroscopy; N.A. and S.A. conducted SEM imaging; T.K. conducted FTIR spectroscopy and compression test; Y. Z. conducted MD simulation and polymorph prediction. B.F. wrote the MATLAB script that was applied to reconstruct the cell migration trajectory. K.A. and A.T. advised the study. X.H. and Y.Z. wrote the paper.

## Competing interests

The authors declare no competing interests.
