## [Peer Review File · Nature Communications]

REVIEWER COMMENTS

Reviewer #1 (Remarks to the Author):

The authors used a bottom-up nanofabrication strategy to manipulate ligand density and determine its effect on cancer cell migration. The authors used IKLLI from laminin alpha 1 chain as the assembling ligands. To facilitate intermolecular aromatic interaction, the authors coupled repeats of L-phenylalanine (F, FF, and FFF) to the N-terminus of IKLLI. The authors found that FFF-IKLLI treatment suppressed cancer cell migration without inducing cytotoxicity. There are several major and minor comments below:

(Major comments)

1. The authors did a lot of works to show the treatment of FFFIKLLI inhibits cancer cell migration through integrin $\alpha 3\beta 1$ and Rac1 activity, but how can FFFIKLLI regulate cellular signals is still unclear. Because all cells can secrete extracellular matrix to allow cells to form integrin-based adhesions and attach to the substratum, does FFFIKLLI peptides regulate integrin signaling through cross-reaction with the extracellular matrix, or regulate the integrin activity on the dorsal side of a cell to fully affect the integrin signals from cell-extracellular matrix adhesions?
2. In Figure S1, the results of CD spectra do not support the sentence "FFFIKLLI preserved the conformation of IKLLI". The FFFIKLLI shows random coil secondary structure, while IKLLI does not show the same secondary structure. In addition, using MicroScale Thermophoresis to show integrin binding affinity is not convincing. The authors should use flow cytometry and cell adhesion assay to determine whether FFFIKLLI peptides only target malignant cancer cells, not benign cancer cells, via integrin $\alpha 3\beta 1$, and determine whether FFF peptides reduce the binding affinity of FFFIKLLI peptides to cell surface.
3. The authors add FFFIKLLI and FFF in culture medium to assemble nanofilaments on the surface of cells, does the nanofilaments activate integrin $\alpha 3\beta 1$? The authors should show whether the nanofilaments co-localize with active integrin $\alpha 3\beta 1$ in malignant cancer cells, not benign cancer cells. In addition, does the nanofilaments regulate integrin-mediated signals, including the phosphorylation level of FAK (Y397) and paxillin (Y118)?
4. In Figure 1b, the authors should describe clearly how to generate the estimated molecular packaging structure from TEM images? To demonstrate cell migration, the authors should use random migration assay, and calculate migration speed, velocity, and directional persistence. In addition, what is the time scale in Figure 1e?
5. In Figure 2a, the authors should show the representative images and videos to confirm that the measurement area is within the leading edge, not the trailing edge. In addition, the authors should describe the method of quantifying edge velocity clearly.

6. In the experiments of traction stress, the authors should describe the method clearly. For example, what is the stiffness of PDMS substrates? What kind of extracellular matrix proteins are coated on the PDMS substrate? How to define the area of measurement? How to quantify the average traction stress?

7. Does FFFIKLLI suppress RhoA and Rac1 activity? And, FFF+FFFIKLLI rescue RhoA and Rac1 activity?

8. To confirm the inhibitory effect of FFFIKLLI in contractile force, quantifying the level of p-MLC(18/19) by western blotting is required.

9. The results in Figure 3a does not support the sentence “while the stress-fiber-associated FAs slide inward, the actin cytoskeleton at the cell rear was not fully disassembled.” The authors should show time-lapse images of actin and FAs (ex: paxillin or integrin) in control or treatment with FFFIKLLI and FFF+FFFIKLLI.

10. The images in Figure 3b, 3c, S18, S19 cannot support the sentence in line 194 “we only observed the co-localization of integrin.....at the cell rear while vinculin, paxillin, and FAK localized in the inward-sliding FAs.....”. To clearly show the region of cell rear and inward-sliding FAs, the authors should include the time-lapse images of actin and FAs (ex: vinculin, paxillin, and FAK) in control or treatment with FFFIKLLI and FFF+FFFIKLLI. In addition, the results cannot support the sentence in line 196 “Such segmentation of FA complex which led to failed FA disassembly.....excessive binding interaction between integrin.....via self-assembly”. The authors should examine FA dynamics using time-lapse images in control or treatment with FFFIKLLI and FFF+FFFIKLLI.

11. Vin256 is a mutant that interacts with talin and paxillin, but not actin filaments. However, talin can associate with actin filaments in FA complex. Therefore, the descriptions about Figure 4a and 4b are all wrong. Also, the images of FAs have poor quality.

12. The authors claimed that Rho activator II can rescue the effect of FFFIKLLI in membrane retraction, and Tiam1-induced Rac1 activation can restore the effect of FFFIKLLI in the membrane protrusion. Does FFFIKLLI inhibit the activity of RhoA and Rac1? Does FFFIKLLI regulate different signaling pathways in different regions of a cell? Do the authors quantify membrane protrusion and retraction in the leading edge or trailing edge in Figure 4e and Figure 4h? The authors should show representative images and videos.

13. The authors claimed that constant activation of Rac1/Tiam1 signaling was the only effective rescue from in-side out. In-side out signals mean activation of integrin from intracellular domain of integrin. So, how does Rac1/Tiam1 signals activate integrin in-side out? Does talin or kindlin involve? What is the role of integrin $\alpha 3 \beta 1$ in the cells with Tiam1-mediated Rac1 activation? Does Tiam1-mediated Rac1 activation can regulate FFFIKLLI-mediated integrin signals?

(Minor comments)

1. Figure S1g-i is not shown in Figure S1.

2. Line 71, Figure S4 should be Figure S4 and S5.

3. For the experiments of wound healing, the authors should show the results as “percentage of wound closure”, include representative images, and have statistical analysis.

4. To show the expression of integrin $\alpha 3 \beta 1$ in different cell lines, RT-PCR is required.
5. Line 174, please check if pEGFPC1-mEGFP-paxillin is correct.
6. Line 222, NAs should be FAs.
7. For the TIAM1 membrane translocation, the authors should include the details in the materials and methods.

Reviewer #2 (Remarks to the Author):

This article presents a study about mechanistic understanding of cell migration through engineering integrin ligand assembly. The authors propose a bottom-up fabrication strategy to enhance the resolution of such systems to the molecular level. The manuscript is coherent and well-written. However, there are questions that the reviewer feels need to be investigated and answered before the article's significance can be established:

-The Introduction part of the paper needs more references and clarification on previous work and how this work is different from/improving upon the past work.

-The authors should provide a more in-depth physical insight about how raising the proportion of FFF may lead to the suppression effect on cell migration. For example, what is special about 1 to 44, 1 to 89 or 1 to 249 ratio values that they discuss. Have they performed any simulations or analytical modeling to consider various ranges of the ratios and observe how changing the ratio affects the system dynamics? How are the estimated molecular packing structures obtained for Figure 1b sketches?

-Along the same lines, for quantitative estimation of surface ligands, what kind of initial configuration/setup was used within MD simulations? Is this what Figure S8 is trying to explain? Were the simulations fully atomistic or coarse-grained? Was there a timestep used within the MD simulations or were all simulations performed in molecular statics form for energy minimization purposes? What procedure/measure was used to calculate surface ligand density from the simulation results? Unless there are page/figure-number limitations specified by the journal, I would recommend the authors produce a separate Figure describing the procedure and the results of the simulations.

- As a minor point, in Figure 1b,d,e, the text is very small and almost not readable.

- I find the discussion about the results of Figure 2 somewhat lacking. What do the authors mean with “super high” and “low” ligand densities? Can they clarify/quantify this parameter based on their measurements or any simulations? Can the authors comment on any physical/chemical mechanisms that cause the restriction on the formation of protrusions? Are entropic effects/entropic penalties of any significance during this process?

-As a minor side note, I believe the authors have a typo on line 132: Therefore, to “assess”

-The Discussion section reads more like a “Conclusion” section. I find minimal discussion about the physics behind why the authors observe specific results throughout the manuscript. In the current version, the paper reads as a summary of figures and results, without significant explanations about the mechanisms employed/discovered in this work. The Discussion section needs to be reworked to reflect the novelty of the work more significantly.

REVIEWER COMMENTS

Reviewer #1 (Remarks to the Author):

[The authors used a bottom-up nanofabrication strategy to manipulate ligand density and determine its effect on cancer cell migration. The authors used IKLLI from laminin alpha 1 chain as the assembling ligands. To facilitate intermolecular aromatic interaction, the authors coupled repeats of L-phenylalanine (F, FF, and FFF) to the N-terminus of IKLLI. The authors found that FFF-IKLLI treatment suppressed cancer cell migration without inducing cytotoxicity. There are several major and minor comments below:]

(Major comments)

[1.The authors did a lot of works to show the treatment of FFFIKLLI inhibits cancer cell migration through integrin $\alpha_3\beta_1$ and Rac1 activity, but how can FFFIKLLI regulate cellular signals is still unclear. Because all cells can secrete extracellular matrix to allow cells to form integrin-based adhesions and attach to the substratum, does FFFIKLLI peptides regulate integrin signaling through cross-reaction with the extracellular matrix, or regulate the integrin activity on the dorsal side of a cell to fully affect the integrin signals from cell-extracellular matrix adhesions?]

We thank the reviewer's comments. To address the concerns raised by the reviewer in regard of the possible interferences induced by cell secreted ECM, we conducted a series of experiments, including cell adhesion assay and cell spreading evaluation assay, to examine with or without natural ECM, especially fibronectin (FN) and laminin-5 (LM-5) that bind to integrin $\alpha_3\beta_1$, whether the regulating efficacy of FFFIKLLI on integrin activity was affected. The experimental results were summarized in Figure S9, which indicated that with or without the extracellular matrix, the regulating efficacy of FFFIKLLI on integrin activity was not affected, and FFFIKLLI could fully regulate the integrin activity without cell-ECM adhesions. We added the experimental results in the Supporting Information, and also revised the main text to address the comments.

Addition in the Supporting Information:

Fig. S9.

Cell adhesion assay was performed by seeding HuH-7 cells on FFFIKLLI pre-coated dishes, and treated with IgG (Ctrl), laminin-5 function-inhibitory antibody P3H9-2, and fibronectin-blocking antibody IST-9. Representative images are shown in (a), and the quantitative results are presented in (b). Cell spreading evaluation was performed by treating the HuH-7 cells with IgG (Ctrl), FFFIKLLI/FFF (1/249) and IgG, FFFIKLLI/FFF (1/249) and IST-9, FFFIKLLI/FFF (1/249) and P3H9-2 for 12 hr. Representative images are shown in (c), and the quantitative results are presented in (d). Cell spreading evaluation was performed by seeding HuH-7 cells on glass,

fibronectin (FN) coated dish, LM-5 coated dish, culturing for 4 hr, followed by the treatment of FFFIKLLI for 12 hr. Representative images are shown in (e), and the quantitative results are presented in (f).

Addition:

No matter blocking or establishing the integrin-ECM adhesion, the regulation effect of nanofilaments were not affected (Figure S9), which indicated that FFFIKLLI could fully regulate integrin $\alpha_3\beta_1$ without cell-ECM adhesions.

[2.In Figure S1, the results of CD spectra do not support the sentence "FFFIKLLI preserved the conformation of IKLLI". The FFFIKLLI shows random coil secondary structure, while IKLLI does not show the same secondary structure.]

We agree with the reviewer. We revised the discussion of CD spectra in the main text and cited the related references.

Original:

Preliminary evaluations indicated that FFFIKLLI preserved the conformation of IKLLI (Figure S1a, b), enhanced the integrin-binding affinity (Figure S1c-e), and selectively targeted integrin $\alpha_3\beta_1$ (Figure S1f),(19) a molecular marker of malignant carcinomas, exerting suppression effects on cancer cell migration without inducing cytotoxicity (Figure S1g-i).

Revised:

Compared with IKLLI, all three candidate assembling ligands exhibited enhanced circular dichroism (CD) signals in the far UV region under aqueous condition, which indicated the self-assembly processes led to the formation of supramolecular structures. Both FIKLLI and FFFIKLLI self-assembled into random coil structure, while self-assembly of FFIKLLI formed β -sheet structure (Figure S1a-c). Intriguingly, these assembling ligands all exhibited enhanced integrin-binding affinity. Among them, FFFIKLLI had the highest binding affinity (Figure S1d, e), and selectively targeted integrin $\alpha_3\beta_1$ (Figure S1f),(19) a molecular marker of malignant carcinomas.

[In addition, using MicroScale Thermophoresis to show integrin binding affinity is not convincing. The authors should use flow cytometry and cell adhesion assay to determine whether FFFIKLLI peptides only target malignant cancer cells, not benign cancer cells, via integrin $\alpha_3\beta_1$, and determine whether FFF peptides reduce the binding affinity of FFFIKLLI peptides to cell surface.]

We thank the reviewer's comments. Following the reviewer's suggestion, upon the preliminary characterization and evaluation using MicroScale Thermophoresis, wound-healing assay, and western blotting, we also used flow cytometry and cell adhesion assay to examine FFFIKLLI. And the experimental results were summarized in Figure S4 and F8, which confirmed that FFFIKLLI selectively inhibited malignant cancer cell migration via targeting integrin $\alpha_3\beta_1$, not benign cancer cells lacking integrin $\alpha_3\beta_1$ expression; FFF didn't bind with integrin $\alpha_3\beta_1$; and the

addition of FFF to FFFIKLLI didn't affect the adherence of integrin $\alpha_3\beta_1$ expressing cells on the substrate, but reduced the binding affinity of co-assembled nanofilaments to cell edge.

As reported, we did the preliminary characterization of synthetic molecules' integrin binding affinity using MicroScale Thermophoresis, which indicated that FFFIKLLI selectively bond with integrin $\alpha_3\beta_1$. Following that, we evaluated the integrin binding affinity of FFFIKLLI *in vitro* using wound-healing assay and western blotting. Totally 8 cell lines, including 6 malignant cancer cell lines expressing integrin $\alpha_3\beta_1$, (HuH-7, HeLa, HepG2, A549, MKN1, and U-87 MG), one malignant cancer cell line (MCF-7) and one epithelial cell line (Ect1/E6E7) lacking integrin $\alpha_3\beta_1$ expression, were tested to determine whether FFFIKLLI selectively targets integrin $\alpha_3\beta_1$. We also tested FFFIKLLI on integrin $\alpha_3\beta_1$ -silenced malignant cancer cells. The experimental results were summarized in Figure S2 and S3, which indicated that FFFIKLLI selectively targeted integrin $\alpha_3\beta_1$ to suppress malignant cancer cell migration.

Based on the preliminary results, we selected two malignant cancer cell lines highly expressing integrin $\alpha_3\beta_1$ (HuH-7 and HeLa) and two cell lines lacking integrin $\alpha_3\beta_1$ expression (MCF-7 and Ect1/E6E7) for flow cytometry test and cell adhesion assay. As presented in Figure S4a, flow cytometry experimental results confirmed that both HuH-7 and HeLa cells highly expressed integrin $\alpha_3\beta_1$, while MCF-7 and Ect1/E6E7 cells only expressed integrin β_1 subunit with negligible expression level of integrin α_3 . The cell adhesion experimental results in Figure S4b and S4c indicated that cells expressing integrin $\alpha_3\beta_1$ remained adhered to the FFFIKLLI coated substrate after washing, while cells lacking integrin $\alpha_3\beta_1$ expression did not. Knockdown of the α_3 or β_1 integrin subunit of HuH-7 cells significantly reduced their adherence to the FFFIKLLI coated substrate (Figure S4d and S4e). These experimental results confirmed that FFFIKLLI selectively inhibit malignant cancer cell migration via targeting integrin $\alpha_3\beta_1$, not benign cancer cells lacking integrin $\alpha_3\beta_1$ expression.

FFF was also examined using both MicroScale Thermophoresis and wound-healing assay of HuH-7 cells. The experimental results that were summarized in Figure S1d and S2c indicated that FFF didn't bind with integrin $\alpha_3\beta_1$ nor exert any suppression effect on malignant cell migration. Therefore, FFF was applied as non-functional assembling motif for co-assembly with FFFIKLLI. Following the reviewer's suggestion, we also conducted cell adhesion assay using HuH-7 cells to examine the binding affinity of FFF and the mixture of FFF/FFFIKLLI. As presented in Figure S8a and S8b, FFF had no effect on the adhesion of HuH-7 cells. The addition of FFF to FFFIKLLI did not affect the HuH-7 cell adhesion suggesting that FFF did not reduced the cell adhesion on FFFIKLLI. The SEM images of HuH-7 cells upon the treatment of nanofilaments (Figure 1c and S10) revealed that co-assembled nanofilaments with higher proportion of FFF barely attached to the cell edge, which suggested FFF reduced the binding affinity of FFFIKLLI to the cell edge.

To address the reviewer's comments, we added the experimental results in the Supporting Information, and revised the main text for a clear statement.

Addition in the Supporting Information:

Fig. S4.

(a-b): mRNA expression of integrin β_1 and integrin α_3 in different cell lines by quantitative RT-PCR analysis. Kruskal-Wallis with Dunn's multiple comparisons test was used for analysis of the data. Error bars represent s.d.. (c): Protein expression of integrin β_1 and integrin α_3 in different cell lines by western blotting analysis. (d): Cell surface protein expression of integrin β_1 and integrin α_3 in HuH-7, HeLa, MCF-7 and Ect1/E6E7 cell lines by flow cytometry analysis. (e-f): Representative images and quantitative analysis showing attached HuH-7 cells in plates coated with or without 100 μ M FFFIKLLI. Scale bar represents 100 μ m. n = 30 for each group. Mann-Whitney test was used for analysis of the data. Error bars represent s.d.. (g-i): Knockdown efficiency of integrins in HuH-7. (j, k): 48 hr wound healing rate of peptide FFFIKLLI treatment on integrin knockdown HuH-7 cells. Peptide concentration, 100 μ M. Scale bar represents 100 μ m. Kruskal-Wallis with Dunn's multiple comparisons test was used for analysis of the data. Error bars represent s.d.. (l, m): Representative images and quantitative analysis showing attached HuH-7 cells in plates coated with 100 μ M FFFIKLLI. Scale bar represents 100 μ m. n = 30 for each group. Kruskal-Wallis with Dunn's multiple comparisons test was used for analysis of the data. Error bars represent s.d..

Fig. S5.

Representative images and quantitative analysis showing attached HuH-7 cells in plates coated with peptide self-assembly or co-assembly. The concentration of FFFIKLLI is kept 100 μ M in every mixture. Scale bar represents 100 μ m. n = 30 for each group. Kruskal-Wallis with Dunn's multiple comparisons test was used for analysis of the data. Error bars represent s.d..

Cell Adhesion Assay

The 96-well microplates wells were coated with peptide assemblies (the concentration of FFFIKLLI was kept at 100 μ M constantly) for 12 hr at 37 $^{\circ}$ C before blocked with the Blocking Buffer (2% BSA, 1mM CaCl₂ and 1mM MnCl₂ in PBS) for 1hr at 37 $^{\circ}$ C. Cells were collected from culture dishes and suspended in the Blocking Buffer containing anti-laminin-5 (P3H9-2, 5 μ g/ml, Chemicon), anti-fibronectin (IST-9, 20 μ g/ml, Abcam) or IgG isotype control (20 μ g/ml, Abcam) before immediately seeded to the coated well (20,000 cells/well) and allowed to incubate at 37 $^{\circ}$ C for 1 hr. The wells were washed for three times with the Blocking Buffer and the phase-contrast images were captured by IncucyteS3 (Essen Bioscience) with a 10x objective.

Original:

As shown in Figure 1a, our strategy is to covalently connect non-functional assembling motif to ECM-derived integrin ligand synthesizing an assembling ligand. Self-assembly of the assembling ligand forms nanofilaments exhibiting super high ligand density. Via introducing the non-functional assembling motifs, co-assembled nanofilaments with precisely controlled ligand densities are produced by varying the proportion of the two components. To put our design into practice, repeats of L-phenylalanine (F, FF, and FFF) facilitating intermolecular aromatic interactions,(16, 17) were coupled to the N-terminus of IKLLI derived from laminin α 1 chain(18) generating candidate assembling ligands. Preliminary evaluations indicated that FFFIKLLI preserved the conformation of IKLLI (Figure S1a, b), enhanced the integrin-binding affinity (Figure S1c-e), and selectively targeted integrin $\alpha_3\beta_1$ (Figure S1f),(19) a molecular marker of malignant carcinomas, exerting suppression effects on cancer cell migration without inducing cytotoxicity (Figure S1g-i). Therefore, FFF and FFFIKLLI were selected as assembling motif and assembling ligand, respectively, for the proposed nanofabrication.

Self-assembly of FFFIKLLI and co-assembly of FFFIKLLI with FFF at various ratios (Figure S2) all formed stable rectangular nanofilaments (~100 nm width, ~100-500 nm length) in water (Figure 1b, S3). Self-assembly of FFFIKLLI selectively targeted cancer cells expressing integrin $\alpha_3\beta_1$, including HuH-7, HeLa, HepG2, A549, MKN1, and U-87 MG cells, to suppress their migration without inducing cytotoxicity (Figure S4). By comparison, it exerted negligible influence on cell lines lacking integrin $\alpha_3\beta_1$ expression, for example MCF-7, Ect1/E6E7 cells (Figure S5a-c), and integrin $\alpha_3\beta_1$ -silenced HuH-7 cells (Figure S5d-g). Co-assembled nanofilaments attenuated the suppression effect on cell migration by raising the proportion of FFF and led to negligible influence when the two components reached 1 to 44 (FFFIKLLI/FFF) ratio (Figure S6). Intriguingly, at 1 to 89 ratio, co-assembled nanofilaments turned to an opposite function exerting promotion effect on cell migration. At the ratio of 1 to 249, more than 1.5 times faster cell migration was detected.

Revised:

As shown in Figure 1a, our strategy is to covalently connect non-functional assembling motif to ECM-derived integrin ligand synthesizing an assembling ligand. The assembling ligands self-assemble into nanofilaments displaying super high ligand density. By mixing the non-functional assembling motif with assembling ligand at different proportions, the co-assembled nanofilaments displaying precisely controlled ligand densities are produced. To put our design into practice, repeats of L-phenylalanine (F, FF, and FFF) facilitating intermolecular aromatic interactions,^(16, 17) were coupled to the N-terminus of IKLLI derived from laminin α_1 chain⁽¹⁸⁾ generating candidate assembling ligands. Compared with IKLLI, all three assembling ligands exhibited enhanced circular dichroism (CD) signals in the far UV region under aqueous condition, which indicated the assembly processes led to the formation of supramolecular structures. Specifically, both FIKLLI and FFFIKLLI self-assembled into random coil structure, while self-assembly of FFIKLLI formed β -sheet structure (Figure S1a-c). Intriguingly, these assembling ligands all exhibited enhanced integrin-binding affinity in solution. Among them, FFFIKLLI had the highest binding affinity (Figure S1d, e), and selectively targeted integrin $\alpha_3\beta_1$ (Figure S1f),⁽¹⁹⁾ a molecular marker of malignant carcinomas. FFF, the non-functional assembling motif exhibited no obvious integrin $\alpha_3\beta_1$ binding affinity (Figure S1d).

To examine the integrin binding affinity in vitro, we tested FFFIKLLI on multiple malignant cancer cell lines expressing integrin $\alpha_3\beta_1$ (HuH-7, HeLa, HepG2, A549, MKN1, and U-87 MG), and cell lines lacking integrin $\alpha_3\beta_1$ expression (MCF-7 and Ect1/E6E7) using wound-healing assay and cell adhesion assay. By adding FFFIKLLI into adhered cell culture, without inducing cytotoxicity (Figure 2a), FFFIKLLI exerted suppression effect on the migration of integrin $\alpha_3\beta_1$ expressing cells (Figure S2b-f), but negligible influence on MCF-7 and Ect1/E6E7 cells (Figure S3). After seeding cells on FFFIKLLI-coated dishes, cells expressing integrin $\alpha_3\beta_1$ remained adhered to the substrate while cells lacking integrin $\alpha_3\beta_1$ expression were washed out (Figure S4a-f). Knockdown of the α_3 or β_1 integrin subunit of HuH-7 cells attenuated the efficacy of FFFIKLLI (Figure 4g-k) and significantly reduced their adherence to the FFFIKLLI coated substrate (Figure S4l and S4m), while knockdown of the α_6 subunit did not. The experimental results confirmed that FFFIKLLI selectively inhibit malignant cancer cell migration via targeting integrin $\alpha_3\beta_1$, not benign cancer cells lacking integrin $\alpha_3\beta_1$ expression.

Consistent to the MicroScale Thermophoresis characterization, FFF didn't affected HuH-7 cell migration or adhesion (Figure S2c, S5), which made it the qualified non-functional assembling motif for our proposed nanofabrication. By fixing the concentration of FFFIKLLI at 100 μ M, we raised the proportion of FFF in a wide range to fabricate co-assembled nanofilaments (Figure S6) displaying different ligand densities for the preliminary evaluation. Self-assembly of FFFIKLLI and co-assembly of FFFIKLLI with FFF formed stable rectangular nanofilaments (~100 nm width, ~100-500 nm length) in water (Figure 1b, S7). Compared with self-assembly, co-assembly gradually attenuated the suppression effect on cell migration by raising the proportion of FFF (Figure S8). When the ratio of FFF to FFFIKLLI reached up to 44:1, co-assembled nanofilaments exhibited negligible influence on HuH-7 cells. Intriguingly, continued growth in the proportion of FFF gradually enhanced the cell migration. When the ratio of FFF to FFFIKLLI reached up to 249, more than 1.5 times faster cell migration was detected.

[3. The authors add FFFIKLLI and FFF in culture medium to assemble nanofilaments on the surface of cells, does the nanofilaments activate integrin $\alpha_3\beta_1$? The authors should show whether the nanofilaments co-localize with active integrin $\alpha_3\beta_1$ in malignant cancer cells, not benign cancer cells. In addition, does the nanofilaments regulate integrin-mediated signals, including the phosphorylation level of FAK (Y397) and paxillin (Y118)?]

We thank the reviewer's comments. Following the suggestion, we examined the nanofilaments and activated integrin subunit β_1 using fluorescent cell imaging. The experimental results were summarized in Figure S16, which indicated that the nanofilaments co-localized with active β_1 integrin subunit on malignant cancer cells expressing integrin $\alpha_3\beta_1$, not cells lacking integrin $\alpha_3\beta_1$ expression. We also evaluated the phosphorylation level of FAK (Y397) and paxillin (Y118) upon the treatment of nanofilaments displaying different ligand densities. The experimental results were summarized in figure S18, which indicated that upon the treatment of FFFIKLLI self-assembled nanofilaments, reduction of FAK Y397 phosphorylation and paxillin Y118 phosphorylation was obtained. Coassembled nanofilaments at 1 to 4 ratio of FFFIKLLI to FFF enhanced the phosphorylation level of both FAK Y397 and paxillin Y118, and a further enhancement was detected when the proportion of FFF was raised to 249.

To address the reviewer's questions, we added the experimental results in the Supporting Information, and revised the main text as below. We also expand the protocol of nanofabrication to clarify the cell treatment procedure. Instead of an *in situ* peptide assembly on the surface of cells upon the treatment, the peptide assembled nanofilaments formed and stabilized before the application on cells.

Addition in the Supporting Information:

Fig. S16.

Immunofluorescence of activated integrin β_1 (12G10) and co-stained with Congo red and phalloidin for each treatment condition. Gray: phalloidin; green: anti-activated integrin β_1 ; magenta: Congo red.

Figure S18.

Protein expression of pY³⁹⁷ FAK, FAK, pY¹¹⁸ Paxillin, Paxillin and GAPDH in HuH-7 cells treated with or without peptide assembly for 12 hr. Kruskal-Wallis with Dunn's multiple comparisons test was used for analysis of the data. Error bars represent s.d..

Original:

Possessing different ligand density, nanofilaments demonstrated a variety of intimacy to the cell edge, especially the finger-like projections. Upon treatment of various peptide assemblies, the super-resolution SEM images of HuH-7 cells exhibited that nanofilaments with super high ligand density almost entangled with all peripheral projections. By reducing the ligand density, only part of the cell edge had an intimate association with nanofilaments leaving more and more cell projections untouched. By reaching low ligand density, the nanofilaments mainly attached to the apical membrane covering microvilli while the whole cell edge was untouched (Figure 1c, S10). Because cell migration is tightly associated with cell morphology, we next investigated the influence of nanofilaments on cell spreading and characterized the correlated

cell motilities (Figure 2). Without treatment, HuH-7 cells were round and less spread on glass (Ctrl). Upon the treatment of nanofilaments with super high ligand densities, restricted HuH-7 cells exhibited reduced spreading area with tentacle-like actin extensions in all directions. By reducing the ligand density from high to intermediate level, HuH-7 cells gradually resumed the smooth cell edge correlated to their partially restored motility (Figure 2a, 2b). Upon the treatment of nanofilaments with low ligand density, cells exhibited broad, flat lamellipodia correlated to almost 2 times enhancement on both maximal traveled distance and total traveled distance (Figure 2c, 2d).

Revised:

Although activated integrin β_1 co-localized with various FFFIKLLI-containing nanofilaments on integrin $\alpha_3\beta_1$ expressing cells (Figure S16), possessing different ligand density, nanofilaments demonstrated a variety of intimacy to the cell edge, especially the finger-like projections (Figure 1c, S17). The correlated influence on the phosphorylation of FAK and paxillin in different manners (Figure S18) suggested a series of regulations of integrin-mediated signals. Upon treatment of various peptide assemblies, the super-resolution SEM images of HuH-7 cells exhibited that nanofilaments with super high ligand density almost entangled with all peripheral projections. Meanwhile, a significant reduction of FAK Y397 phosphorylation and paxillin Y118 phosphorylation was detected. By reducing the ligand density, only part of the cell edge had an intimate association with nanofilaments leaving more and more cell projections untouched. Under the same conditions, elevated levels of phosphorylated FAK and paxillin were obtained. By reaching low ligand density, the nanofilaments mainly attached to the apical membrane covering microvilli while the whole cell edge was untouched. And further enhancement on phosphorylation of FAK and paxillin was achieved.

Because cell migration is tightly associated with cell morphology, we next investigated the influence of nanofilaments on cell spreading and characterized the correlated cell motilities (Figure 2). Without treatment, HuH-7 cells were round and less spread on glass (Ctrl). Upon the treatment of nanofilaments with super high ligand densities, restricted HuH-7 cells exhibited reduced spreading area with tentacle-like actin extensions in all directions. By reducing the ligand density from high to intermediate level, HuH-7 cells gradually resumed the smooth cell edge correlated to their partially restored motility (Figure 2a, 2b). Upon the treatment of nanofilaments with low ligand density, cells exhibited broad, flat lamellipodia correlated to almost 2 times enhancement on both maximal traveled distance and total traveled distance (Figure 2c, 2d).

Original in the Supporting Information:

Preparation of peptides assemblies

To prepare the stock solutions of peptide assemblies, required amount of peptide was dissolved in Milli-Q water at 100 mM (for FFF) or 10 mM (for the other peptides) and adjust the solution to reach a final pH of 7.0 using 1N NaOH solution. To prepare the working solution, the desired amount of stock solution was diluted using Milli-Q water or culture medium. Stand the working solutions for 30 min before applying them to the experiments.

Revised in the Supporting Information:

Preparation of peptides assemblies

To prepare the stocks of peptide assembled nanofilaments, required amount of peptide was dissolved in Milli-Q water at 100 mM (for FFF) or 10 mM (for the other peptides, and peptides mixture) and adjust the solution to reach a final pH of 7.0 using 1N NaOH solution. To stabilize the peptide assemblies, the stock solutions were stored at room temperature for at least 24 hours before application. The obtained nanofilaments were very stable and were not affected by simple dilution at neutral pH. Therefore, to prepare the working solution, the desired amount of stock solution was diluted using Milli-Q water or culture medium. Stand the working solutions for 30 min to evenly distribute nanofilaments in the working solution before applying them to the experiments.

[4.In Figure 1b, the authors should describe clearly how to generate the estimated molecular packaging structure from TEM images?]

Following the reviewer's suggestion, we expanded the content of "Molecular dynamics simulation and polymorph prediction" in the Supporting Information with detailed protocols. Besides that, we also added the step-by-step analysis results in the Supporting Information.

Original:

Molecular dynamics simulation and polymorph prediction

Molecular mechanics calculations were performed using Materials Studio. For all simulations, the Ewald method (3,4) was used for the electrostatic and van der Waals interaction terms. Gasteiger charges were used for an initial conformational search. As the crystal structure prediction method uses a rigid body approximation in the initial search for crystal packing alternatives, the analysis to determine low energy geometry was performed by following the protocols reported by Kim etc. (5), and the results were used as input for the packing calculations. The conformation of FFF was reported by Ellenbogen etc(6). Therefore, FFFIKLLI was drawn based on the structure of the FFF motif, and geometrical energy minimization scans were performed using the Forcite module of Materials Studio. After finding the lowest energy conformation of FFFIKLLI, the reported structure of the FFF unit cell was used as the starting point for crystal structure prediction using the Materials Studio Polymorph Predictor (PP). By replacing FFF by FFFIKLLI from the reported unit cell, PP calculation was performed.

The PP was set to its default fine setting (this sets the simulated annealing algorithm to a temperature range of 300-100000.0 K with a heating factor of 0.025, requiring 12 consecutive steps to be accepted before cooling and a maximum of 7000 steps) with the force field Dreiding 2.21 with Gasteiger charges. The 10 most common space groups found in organic crystals registered in the CSD were selected, including $P2_1/c$, $P1$, $P2_12_12_1$, $P2_1$, $C2/c$, $Pbca$, $Pna2_1$, $Pbcn$, Cc , and $C2$. Clustering of the predicted polymorphs was done using the polymorph clustering routine in Materials Studio. After the final clustering, hydrogen bonding analysis was performed on the calculated crystal structures to identify the packing modes matching the FTIR spectra regarding the hydrogen bonding signals. After extending the structure along the unit axes, the surface that exposes most integrin ligand IKLLI was presented in a defined square area aligned to the self-assembled nanostructures. The

molecular packing of the mixture of FFFIKLLI and FFF at 1:249 ratio was predicted based on the crystal structure of FFF unit cell and the molecular packing structure of 1:4 ratio. The ligand distance was measured using the distance measurement functions of Materials Studio.

Revised:

Molecular dynamics simulation and polymorph prediction

Molecular mechanics calculations were performed using Materials Studio® 2020. For all simulations, the Ewald method (3,4) was used for the electrostatic and van der Waals interaction terms. Gasteiger charges were used for an initial conformational search. As the crystal structure prediction method uses a rigid body approximation in the initial search for crystal packing alternatives, the analysis to determine low energy geometry was performed by following the protocols reported by Kim etc. (5), and the results were used as input for the packing calculations.

The conformation of FFF was reported by Ellenbogen etc(6). As reported, the single crystal structure of FFF in FFF-tape (self-assembled nanostructure) was determined by single crystal XRD measurements to 1.1 Å resolution. The determined structure is triclinic, space group P1, with four FFF molecules per asymmetric unit. The alignment of the unit cell with regards to the self-assembled tape structure reveals that growth is governed by π -interactions between adjacent aromatic rings along the c-axis.

1. Import the crystal structure of FFF to Materials Studio and draw FFFIKLLI based on the conformation of FFF. The geometrical energy minimization scans were performed using the Forcite module of Materials Studio. The molecule in Fig. S8a was found to have the lowest energy conformation. The force field used was Dreiding 2.21 with Gasteiger charges as implemented in the Materials Studio packages.
2. Import the FFF unit cell obtained via single crystal XRD measurements as described in the protocol to Materials Studio. Replacing FFF by optimized FFFIKLLI to 1:1, 2:1 and 4:1 ratio, respectively. The initial placement of FFFIKLLI was determined by π -interactions between adjacent aromatic rings along the c-axis. The geometrical energy minimization scans were performed using the Forcite module. The optimized gas phase conformations as presented in Fig. S11 were used as the starting points for crystal structure prediction using the Materials Studio Polymorph Predictor (PP).
3. The PP was set to its default fine setting (this sets the simulated annealing algorithm to a temperature range of 300-100000.0 K with a heating factor of 0.025, requiring 12 consecutive steps to be accepted before cooling and a maximum of 7000 steps) with the force field Dreiding 2.21 with Gasteiger charges. The 10 most common space groups found in organic crystals registered in the CSD were selected, including P2₁/c, P1, P2₁2₁2₁, P2₁, C2/c, Pbcn, Pna2₁, Pbcn, Cc, and C2. Clustering of the predicted polymorphs was done using the polymorph clustering routine in Materials Studio. After the final clustering, hydrogen bonding analysis (as implemented in the Materials Studio packages) was performed on the calculated crystal structures (Fig. S12). According to the FTIR spectra, self-assembled FFF presents NH-O hydrogen bonding, self-assembled FFFIKLLI presents NH-N hydrogen bonding, co-assembled FFF and FFFIKLLI present both NH-O and NH-N hydrogen bonding. The reported FFF unit cell shows both intermolecular and intramolecular NH-O hydrogen bonding which

matches to the FTIR results. Based on the summarized hydrogen bonding analysis of the calculated crystal structures (Table S1), we highlighted the structures that match to the FTIR results in black frames.

4. The TEM images and SEM images of self-assembled FFF, self-assembled FFFIKLLI, co-assembled FFF with FFFIKLLI demonstrated that the assembled nanofilaments all shared similar morphologies that were barely influenced by the variation of components' proportion. Taking advantage of unified space group symmetry to reduce the number of variables in searches of molecular packing modes in nanofilaments assembled by FFF and FFFIKLLI at various proportions, we selected the clustering results of Pbcn space group (Fig. S13) which generated matching structures at different proportions to ease the comparisons in regard of ligand (IKLLI) distribution density.
5. After extending the structure along the unit axes, the surface that exposes most integrin ligand IKLLI was presented in a defined square area (10 x10 nm²) within the dimension range of nanofilaments (Fig. S14). The ligand density on the surface of nanofilaments formed by co-assembly of FFFIKLLI and FFF at 1:44, 1: 89, and 1:249 ratio was estimated statistically based on the crystal structure of FFF unit cell by replacing one FFF with FFFIKLLI on the filament that composed of 45, 90, and 250 FFF, respectively. The detailed calculation results are presented in Fig. S15.
6. The exposed ligand IKLLI was identified as effective ligand, and the distance between the effective ligands was measured using the distance measurement functions of Materials Studio by calculating the distance between the C-terminals of the effective ligands (Fig. S14).

Addition in the Supporting Information:

a

b

Figure S11. The molecular and geometry-optimized structures of FFFIKLLI (a), FFF and FFFIKLLI at 1:1, 1:2, and 1:4 ratio (b) presented in stick model.

d

Fig. S12.

Predicted crystal structure unit cell of FFFIKLLI (a), FFFIKLLI:FFF = 1:1 (b), FFFIKLLI:FFF = 1:2 (c), FFFIKLLI:FFF = 1:4 (d), with the $P2_1/C$, $P1$, $P2_12_12_1$, $P2_1$, $C2/c$, $Pbca$, $Pna2_1$, $Pbcn$, Cc , and $C2$ space group symmetry. The structures are presented in stick model.

Table S1. Hydrogen bonding analysis of predicted crystal structure unit cells. Intra represent intramolecular hydrogen bonding and inter represent intermolecular hydrogen bonding.

Space group	FFF		FFFIKLLI:FFF = 1: 4		FFFIKLLI:FFF = 1: 2		FFFIKLLI:FFF = 1: 1		FFFIKLLI	
	NH-O	NH-N	NH-O	NH-N	NH-O	NH-N	NH-O	NH-N	NH-O	NH-N
XRD intra	✓									
XRD inter	✓									
C2-C intra			✓		✓	✓	✓	✓	✓	
C2-C inter					✓					
C2 intra			✓		✓	✓	✓	✓	✓	
C2 inter			✓						✓	
CC intra			✓		✓	✓	✓	✓	✓	
CC inter					✓					
P-1 intra			✓		✓		✓		✓	
P-1 inter				✓			✓			
P21-C intra			✓		✓	✓	✓			
P21-C inter			✓	✓	✓					
P21 intra			✓		✓	✓	✓		✓	✓
P21 inter			✓	✓	✓	✓	✓			
P212121 intra			✓	✓	✓		✓		✓	✓
P212121 inter			✓	✓	✓				✓	
PBCA intra			✓		✓	✓	✓			✓
PBCA inter			✓		✓	✓	✓		✓	
PBCN intra			✓		✓	✓	✓	✓		✓
PBCN inter				✓	✓		✓			
PNA21 intra			✓		✓		✓		✓	✓
PNA21 inter						✓	✓		✓	✓

Fig. S15. (a) Space-filling model of crystal structure unit cell of FFF at zy, zx, xy plain. FFF motif was presented in pink. (b) Because the C-terminus of FFF that can be covalently linked with IKLLI is only exposed toward the zx plain (the image in the middle), we took the area size of this plain for surface calculation. The calculation results of ligand density for FFFIKLLI/FFF at 1 to 249 to 1 to 44 were summarized. (c) The scheme represents the estimation of ligand distance in regard of three different densities.

[To demonstrate cell migration, the authors should use random migration assay, and calculate migration speed, velocity, and directional persistence. In addition, what is the time scale in Figure 1e?]

To demonstrate cell migration, we did both wound healing assay and random migration assay. Wound healing assay was mainly applied for the preliminary evaluations. For example, in Figure S2, S3, S4, and S8, wound healing assay was used to examine the influence of peptide assemblies on cell migration. While, in Figure 2, 3, and 5, random migration assay has been applied for quantitative analysis.

Based on the random migration assay results, we calculated the migration speed, velocity, and directional persistence, and the details were presented in revised Figure 2, Figure S31, S32, S34, and the supporting information.

Revised Figure:

Figure 2. Nanofilaments with various ligand presentations regulate both cell shape and cell migration. (a) The phalloidin staining of Huh-7 cells with and without the treatment of various nanofilaments. Scale bars represent 20 μm . (b) The spreading area and the perimeter area ratio of HuH-7 cells under various conditions. Kruskal-Wallis with Dunn's multiple comparisons test was used for analysis of the data. Error bars represent standard deviation. From left to right, $n = 61, 49, 45, 46, 51, 56$ cells, respectively. The trajectory plots (c), and the correlated quantitative analysis of travel speed, persistence and persistence index (d) of randomly selected migrating cells for each incubation condition. Live cell images were taken every 10 min for a total of 10 hr. Kruskal-Wallis with Dunn's multiple comparisons test was applied in

data analysis. Error bars represent standard error of mean. From left to right, n = 261, 280, 230, 260, 214, and 278 cells, respectively. Scale bars in panel c represent 50 μm .

Fig. S31. Vin258 maintained the FAs on the periphery edge.

(a). Immunofluorescence of pMLC or FAK co-stain with phalloidin. HuH-7 cells were transfected with Pefgpc1/Gg Vcl 1-258 and treated with FFFIKLLI for 12 hr. Scale bar, 20 μm .
(b-c). Spreading area and P/A ratio of vin258-transfected HuH-7 cells with or without treating

with FFFIKLLI for 12 hr. Mann-Whitney test was used for analysis of the data. Error bars represent standard deviation. n = 51 cells. **(d-e)**. Violin plot of all protrusion or retraction velocity values collected from each time point for transfected HuH-7 cells pretreated with or without FFFIKLLI for 12 hr. Mann-Whitney test was used for analysis of the data. Median and quartiles were presented in the plot. n = 10 cells for each group. **(f-h)**. The migration speed directional persistence and persistence index of randomly selected migrating transfected HuH-7 cells. Mann-Whitney test was used for analysis of the data. Error bars represent standard deviation. n= 169, 176 cells, respectively.

Fig. S32. Rho activations reboot trailing tail retraction.

(a). Time-lapse imaging of Rho-preactivated HuH-7 cells (mRuby-Lifeact-7 transfected) upon the treatment of FFFIKLLI. Scale bar, 20 μm . **(b).** Immunofluorescence of FAK and pMLC co-stain with phalloidin. Cells were preincubated with Rho Activator II and treated with FFFIKLLI for 12 hr. Scale bar, 20 μm . **(c, d).** Spreading area and P/A ratio of Rho-preactivated cells with or without treating with FFFIKLLI for 12 hr. Mann-Whitney test was used for analysis of the data. Error bars represent standard deviation. $n = 85, 76$ cells, respectively. **(e-g).** The migration speed, directional persistence and persistence index of randomly selected migrating Rho-preactivated HuH-7 cells. Mann-Whitney test was used for analysis of the data. Error bars represent standard deviation. $n = 219, 221$ cells, respectively. **(h).** Violin plot of all protrusion or retraction velocity values collected from each time point for Rho-activated HuH-7 cells and treated with or without FFFIKLLI for 12 hr. Mann-Whitney test was used for analysis of the data. Median and quartiles were presented in the plot. $n = 10$ cells for each group.

Fig. S34. Tiam1/Rac1 activation reboots the cell migration.

(a). Immunofluorescence of FAK and pMLC co-stain with phalloidin. Tiam1/Rac1 was preactivated right before starting the 12 hr treatment of FFFIKLLI. Scale bar, 20 μm . **(b, c).** Spreading area and P/A ratio of Rho-preactivated cells with or without treating with FFFIKLLI for 12 hr. Mann-Whitney test was used for analysis of the data. Error bars represent standard

deviation. $n = 64$ cells. **(d-f)**. The migration speed, directional persistence time and persistence index of randomly selected migrating Tiam1/Rac1-activated HuH-7 cells. Mann-Whitney test was used for analysis of the data. Error bars represent standard deviation. $n = 230, 234$ cells, respectively. **(g)**. Violin plot of all protrusion or retraction velocity values collected from each time point for Tiam1/Rac1-activated HuH-7 cells and treated with or without FFFIKLLI for 12 hr. Mann-Whitney test was used for analysis of the data. Median and quartiles were presented in the plot. $n = 10$ cells for each group.

Addition in the Supporting Information:

Random Cell Migration Assay

HuH-7 cells were labeled with fluorescent protein fusion constructs or Hoechst 33342. After being treated with or without the peptides for 12 hr, the cells were tracked with a 20x/0.8 Plan-Apochromat objective for 6 hr. The centroids of labeled cells were tracked using the TrackMate plugin (<https://imagej.net/plugins/trackmate/>) in ImageJ(13). A custom MATLAB script generated by Mr. B. Feng was applied to reconstruct the cell migration trajectory. The directionality of the cell movement was described by the persistence index and the persistent time, the persistence index was defined by the ratio of the vectorial distance (the distance between the origin and the endpoint of the movement) and the length of total path and the persistent time is defined as the time it takes for the cell to change the initial direction by 90° . (14)

The trajectory plots in Figure 1e were obtained via time lapse live cell imaging. Images were taken every 10 mins for a total period of 10 hr. We added the information in the revised caption. Original:

Figure 1. Engineering integrin ligand assembly to control ligand presentation. (a) Schematic illustration of precise control of integrin ligand presentation on nanofilaments via peptide assembly. (b) TEM images of nanofilaments obtained via molecular self-assembly and co-assembly of FFFIKLLI (100 μM) and FFF at various ratios, and the estimated molecular packing structures. IKLLI motif is presented in blue and FFF motif is presented in pink. The scale bars represent 200 nm. (c) Zoom-in SEM images (false color) of HuH-7 cell edge and apical membrane after 3-day incubations. FFFIKLLI was maintained at a concentration of 100 μM . Cell body is highlighted in pink, while the nanofilaments are highlighted in blue. The scale bars represent 300 nm. (d) The phalloidin staining of Huh-7 cells. Scale bars represent 20 μm . The trajectory plots (e), and the correlated quantitative analysis of maximal and total travel distance (f) of randomly selected migrating cells for each incubation condition. Kruskal-Wallis with Dunn's multiple comparisons test was applied in data analysis. Error bars represent standard error of mean. $n = 261, 280, 230, 260, 214,$ and 278 cells from left to right panel, respectively. Scale bars in panel e represent 50 μm .

Revised:

Figure 2. Nanofilaments with various ligand presentations regulate both cell shape and cell migration. (a) The phalloidin staining of Huh-7 cells with and without the treatment of various nanofilaments. Scale bars represent 20 μm . (b) The spreading area and the perimeter area ratio of HuH-7 cells under various conditions. Kruskal-Wallis with Dunn's multiple comparisons test was used for analysis of the data. Error bars represent standard deviation. From left to right, $n = 61, 49, 45, 46, 51, 56$ cells, respectively. The trajectory plots (c), and the correlated quantitative analysis of travel speed, persistence and persistence index (d) of randomly selected migrating cells for each incubation condition. Live cell images were taken every 10 min for a total of 10 hr. Kruskal-Wallis with Dunn's multiple comparisons test was applied in data analysis. Error bars represent standard error of mean. From left to right, $n = 261, 280, 230, 260, 214,$ and 278 cells, respectively. Scale bars in panel c represent 50 μm .

[5. In Figure 2a, the authors should show the representative images and videos to confirm that the measurement area is within the leading edge, not the trailing edge. In addition, the authors should describe the method of quantifying edge velocity clearly.]

In Figure 2a, the measurement area is entire periphery of a cell. Upon the treatments by peptide assemblies of FFFIKLLI/FFF at a ratio of 1:0, 1:1, 1:2, and 1:4, the cells exhibited unpolarized morphology without clearly defined leading edge nor trailing edge. Consistent to Figure 2a, the Figure 2b represents the velocity of entire periphery. To clarify that, we revised the main text and figure caption as below. The representative images were added in the Supporting Information as Figure S19.

Original:

Therefore, to access the impact of nanofilaments on migration-correlated membrane dynamics, we performed a morphodynamical analysis by mapping the protrusion and retraction in response to various nanofilaments over time (Figure 2a, S12).

Revised:

Therefore, to assess the impact of nanofilaments on migration-correlated membrane dynamics, we performed a morphodynamical analysis by mapping the protrusion and retraction along the entire periphery in response to various nanofilaments over time (Figure 3a, S19).

Original:

Figure 2. Nanofilaments regulate membrane dynamics and FA organizations in a ligand density dependent manner. Kymograph of normalized edge velocity (a) and mean velocity over time for protrusions and retractions of HuH-7 cells (b), merged F-actin phalloidin staining, and paxillin immunofluorescence in HuH-7 cells (c), heat-scale plots of traction stress magnitudes of HuH-7 cells (d), upon the treatment of various peptide assemblies for 12 hr. Scale bars in panel c, and d represent 5 μm and 20 μm , respectively.

Revised:

Figure 3. Nanofilaments regulate membrane dynamics and FA organizations in a ligand density dependent manner. Kymograph of normalized edge velocity (a) and mean velocity over time for protrusions and retractions along the entire periphery of HuH-7 cells (b), merged F-actin phalloidin staining, and paxillin immunofluorescence in HuH-7 cells (c), heat-scale plots of traction stress magnitudes of HuH-7 cells (d), upon the treatment of various peptide assemblies for 12 hr. Scale bars in panel c, and d represent 5 μm and 20 μm , respectively.

Addition in the Supporting Information:

Fig. S19.

Time-lapse images of mRuby-Lifect-7 transfected HuH-7 cells with the treatment of nanofilaments for 12 hr. Scale bar, 20 μm .

[6. In the experiments of traction stress, the authors should describe the method clearly. For example, what is the stiffness of PDMS substrates? What kind of extracellular matrix proteins are coated on the PDMS substrate? How to define the area of measurement? How to quantify the average traction stress?]

In this research, PDMS substrates without ECM coating were used. And because mRuby-Lifeact-7 transfected HuH-7 cells were used in the experiments, the in situ fluorescent images of HuH-7 cells were applied to define the area of measurement. The correlated fluorescent images of cells were presented in Figure S16. Each single spot in the average traction stress plot represents the mean traction force of the imaged cell. Following the reviewer's suggestions, we input the details regarding the stiffness of PDMS substrates in the Supporting Information, and revised the protocol and the caption of Figure S16 (S23 after revision) for a clearer presentation.

Original:

Traction Stress analysis

mRuby-Lifeact-7 transfected HuH-7 cells were seeded on a PDMS substrate functionalized with 0.2 μm FluoSpheres™ Carboxylate-Modified Microspheres (F8807, Invitrogen) at a density of approximately 6000 cells / cm^2 . After being pretreated with peptides for 12 hr, the samples were imaged by a 20x/0.8 Plan-Apochromat objective, and the bead displacement obtained from confocal imaging was converted into force-displacement fields following established protocol(13, 14). Briefly, Images of beads with and without cell attachment were first aligned to correct experimental drift using ImageJ plugin "align slices in stack". The displacement field was subsequently calculated by another ImageJ plugin "PIV (Particle Image Velocimetry)". The cross-correlation PIV with 64 \times 32-pixel size was used on all images for the PIV analysis to produce the position and vector field of the bead displacement. With the displacement field obtained from the PIV analysis, the traction force field was then reconstructed by the ImageJ plugin "FTTC".

Revised:

Traction Stress analysis

mRuby-Lifeact-7 transfected HuH-7 cells were applied for the traction force microscopy experiments. The in situ fluorescent image of the cell was applied to define the area of measurement. The cells were seeded on a PDMS substrate functionalized with 0.2 μm FluoSpheres™ Carboxylate-Modified Microspheres (F8807, Invitrogen) at a density of approximately 6000 cells / cm^2 . The substrate was fabricated by following the published protocol on (STAR)Protocol (14), except that there were no ECM proteins coated on the substrate here. The stiffness of the PDMS substrate was characterized using compression test, and the results were summarized below which indicated that the Young's modulus of the PDMA substrate was 12.1 ± 0.29 KPa. After being pretreated with peptides for 12 hr, the samples were imaged by a 100x/1.35 Silicon UPlanSApo objective, and the bead displacement obtained from confocal imaging was converted into force-displacement fields following established protocol(16, 17). Briefly, Images of beads with and without cell attachment were first aligned to correct experimental drift using ImageJ plugin "align slices in stack". The displacement field was subsequently calculated by another ImageJ plugin "PIV (Particle Image

Velocimetry)". The cross-correlation PIV with a size of 64×32 pixel was used on all images for the PIV analysis to produce the position and vector field of the bead displacement. With the displacement field obtained from the PIV analysis, the traction force field was then reconstructed by the ImageJ plugin "FTTC". The correlated lifeact images were used to define the cell area and the traction boundary of the stress.

Original:

Fig. S16. Peptide assemblies with various ligand densities control the cell traction force.

(a) Morphology of HuH-7 cells cultured on PDMS substrates. Cell was transfected with mRuby-Lifeact-7 and treated with peptide assemblies for 12 hr. Scale bar, 20 μm. (b-c) Average and Maximal traction stress of each condition. Kruskal-Wallis with Dunn's multiple comparisons test was used for analysis of the data. Error bars represent standard deviation. n = 20 (Ctrl), 18 (1:0), 19 (1:1), 20 (1:2), 21 (1:4), 21 (1:249) cells.

Revised:

Fig. S23. Peptide assemblies with various ligand densities control the cell traction force.

(a) Morphology of HuH-7 cells cultured on PDMS substrates. Cell was transfected with mRuby-Lifeact-7 and treated with peptide assemblies for 12 hr. Scale bar, 20 μm. (b-c) Average and Maximal traction stress of each condition. The average traction stress represents the mean traction force of each cell. Kruskal-Wallis with Dunn's multiple comparisons test was used for analysis of the data. Error bars represent standard deviation. n = 20 (Ctrl), 18 (1:0), 19 (1:1), 20 (1:2), 21 (1:4), 21 (1:249) cells.

[7.Does FFFIKLLI suppress RhoA and Rac1 activity? And, FFF+FFFIKLLI rescue RhoA and Rac1 activity?]

The FFFIKLLI self-assembled nanofilaments suppressed Rac1 activity that was confirmed via live-cell imaging of a FRET-based Rac1 biosensor. The experimental results were presented in

Figure 4d, 4e by following the published protocol (*Mol. Biol. Cell* 2011, 22, 4647). The correlated information was presented in the Supporting Information.

To fully address the reviewer's comments, we examined the Rac1 activity upon the treatment of FFF/FFFIKLLI co-assemblies, and the RhoA activity upon the treatment of self-assembly of FFFIKLLI and FFF/FFFIKLLI co-assemblies. The experimental results were summarized in Figure S30. And the results indicated that self-assembled FFFIKLLI suppressed RhoA activity, FFF/FFFIKLLI co-assemblies not just rescued but also enhanced RhoA and Rac1 activity by increasing the proportion of FFF. We also revised the main text as below.

Addition in the Supporting Information:

Fig. S30.

Representative FRET/CFP ratio images and quantitative analysis of HuH-7 cells expressing RaichuEV-Rac1 or DORA-RhoA with or without the treatment of peptide assemblies for 12 hrs. The images were coded according to a pseudo color scale, which ranges from yellow to purple with an increase in FRET activity. Scale bars represent 20 μ m. n=50 cells for each group. Kruskal-Wallis with Dunn's multiple comparisons test was used for analysis of the data. Error bars represent s.d..

Original:

Together with a great inhibition of Rac1 activity (Figure 4d) on cell periphery (Figure 4e) which indicated the prevention of both protrusion formation and forward motion, it was demonstrated that self-assembly of FFFIKLLI restricted both trailing edge retraction and leading-edge protrusion of HuH-7 cells resulting into depolarization suppressing cell motility.

Revised:

Together with a great inhibition of Rac1 and RhoA activity on cell periphery (Figure 4d, 4e, and S30) which indicated the prevention of both protrusion formation and forward motion, it was demonstrated that self-assembly of FFFIKLLI restricted both trailing edge retraction and leading-edge protrusion of HuH-7 cells resulting into depolarization suppressing cell motility.

[8.To confirm the inhibitory effect of FFFIKLLI in contractile force, quantifying the level of p-MLC(18/19) by western blotting is required.]

We thank the reviewer's suggestion. The western blotting experiment was conducted and the results were summarized in Figure S24 added in the Supporting Information, and addressed in the main text.

Addition in the Supporting Information:

Fig. S24.

Protein expression of Phospho-Myosin Light Chain 2 (Thr18/Ser19) (pMLC) and GAPDH in HuH-7 cells treated with or without peptide assembly for 12 hr. Kruskal-Wallis with Dunn's multiple comparisons test was used for analysis of the data. Error bars represent s.d..

Original:

Compared with control, treated HuH-7 cells exhibited more than 60% reduction on traction stresses (Figure 3d and Figure S22).

Revised:

Compared with control, treated HuH-7 cells exhibited more than 60% reduction on traction stresses (Figure 3d and Figure S23), which was consistent to a dramatic decrease of pMLC expression (Figure S24).

[9. The results in Figure 3a does not support the sentence "while the stress-fiber-associated FAs slide inward, the actin cytoskeleton at the cell rear was not fully disassembled." The authors should show time-lapse images of actin and FAs (ex: paxillin or integrin) in control or treatment with FFFIKLLI and FFF+FFFIKLLI.]

Following the reviewer's suggestion, we added the time-lapse images of actin and FAs, actin and integrin, actin and paxillin in the Supporting Information, and cite the figure in the main text.

Addition in the Supporting Information:

Fig. S25.

Time-lapse images showing the dynamics of actin cytoskeleton (grey) and focal adhesion proteins (green) in HuH-7 cells co-transfected with pGFP-FAK upon the treatment of peptide assemblies for 12 hrs. Scale bar, 5μm.

Fig. S26.

Time-lapse images showing the dynamics of actin cytoskeleton (grey) and focal adhesion proteins (red) in HuH-7 cells co-transfected with mRuby-Lifeact-7 and mVenus-Integrin-Beta1 or EGFP-talin upon the treatment of peptide assemblies for 12 hrs. Scale bar, 5 μ m.

Fig. S27.

Time-lapse images showing the dynamics of actin cytoskeleton (grey) and focal adhesion proteins (green) in HuH-7 cells co-transfected with mRuby-Lifeact-7 and mGFP-paxillin or pEGFP-vinculin upon the treatment of peptide assemblies for 12 hrs. Scale bar, 5 μm.

Original:

By tracking actin and FA dynamics via time-lapse imaging of mRuby-Lifeact and EGFP-paxillin co-transfected HuH-7 cells (Figure S24), we observed that while the stress-fiber-associated FAs slide inward, the actin cytoskeleton at the cell rear was not fully disassembled (Figure 4a).

Revised:

By tracking actin and FA dynamics via time-lapse imaging of mRuby-Lifeact and mEGFP-paxillin co-transfected HuH-7 cells, we observed that while the stress-fiber-associated FAs slide inward, the actin cytoskeleton at the cell rear was not fully disassembled (Figure 4a, S25-27).

[10. The images in Figure 3b, 3c, S18, S19 cannot support the sentence in line 194 " we only observed the co-localization of integrin.....at the cell rear while vinculin, paxillin, and FAK localized in the inward-sliding FAs.....". To clearly show the region of cell rear and inward-sliding FAs, the authors should include the time-lapse images of actin and FAs (ex: vinculin, paxillin, and FAK) in control or treatment with FFFIKLLI and FFF+FFFIKLLI. In addition, the results cannot support the sentence in line 196 "Such segmentation of FA complex which led to failed FA disassembly.....excessive binding interaction between integrin.....via self-assembly". The authors should examine FA dynamics using time-lapse images in control or treatment with FFFIKLLI and FFF+FFFIKLLI.]

We thank the reviewer's comments. Following the suggestions, we added the time-lapse images of actin and FAs, actin and integrin, actin and paxillin, actin and vinculin in the Supporting Information, and cite the figures in the main text. Please refer to the response to the comment above for the figures.

Original:

Different from the well-studied reduced trailing edge retraction caused by stable adhesion within the cell rear,(27) we only observed the co-localization of integrin $\alpha_3\beta_1$ with talin and α -actinin remaining on the actin filaments at the cell rear while vinculin, paxillin, and FAK located in the inward-sliding FAs connected with stress fibers (Figure 4b,4c, S25, S26). Such segmentation of FA complex which led to failed FA disassembly on cell edge is highly possible due to the excessive binding interaction between integrin $\alpha_3\beta_1$ and ligands clustered in a super high density via self-assembly.

Revised:

Different from the well-studied reduced trailing edge retraction caused by stable adhesion within the cell rear,(27) we only observed the co-localization of integrin $\alpha_3\beta_1$ with talin (Figure S26) and α -actinin remaining on the actin filaments at the cell rear while vinculin, paxillin, and FAK located in the inward-sliding FAs connected with stress fibers (Figure 4b,4c, S25, S27). Such segmentation of FA complex which led to failed FA disassembly on cell edge is highly possible due to the excessive binding interaction between integrin $\alpha_3\beta_1$ and ligands clustered in a super high density via self-assembly.

[11. Vin256 is a mutant that interacts with talin and paxillin, but not actin filaments. However, talin can associate with actin filaments in FA complex. Therefore, the descriptions about Figure 4a and 4b are all wrong. Also, the images of FAs have poor quality.]

Following the reviewer's comment, we corrected the descriptions about Figure 4a and 4b. We also improved the image quality.

Original:

By expressing vin258, a mutant that possesses vinculin D1 domain exhibiting high affinity to talin and paxillin but lack of actin-binding domain,(26) in HuH-7 cells, FA complex maintained united on the periphery without connecting to stress fibers after 12 hr treatment of FFFIKLLI (Figure 4a, S20a). However, expressing the FA stabilizing forms of vinculin failed to preserve the actomyosin network, could not resume protrusion nor trailing edge retraction (Figure 4b, S20b-e). Eventually, the suppression effect on cell motility was remained (Figure 4c, S20f-g). To preserve FAs on cell edge and associated with stress fibers, we applied Rho Activator II on HuH-7 cells to drive elevated level of actomyosin contractility.(27, 28)

Revised:

In regard of the essential role of vinculin in the regulation of the assembly and disassembly of adhesion receptor complexes, we expressed vin258, a mutant that possesses vinculin D1 domain exhibiting high affinity to talin and paxillin but lack of actin-binding domain,(26) in HuH-7 cells, to maintain united FA complex on the periphery even upon a 12 hr treatment of FFFIKLLI (Figure 5a, S31a). However, solely allocating vinculin in FAs but without actomyosin-mediated forces directly acting on vinculin could not resume protrusion nor trailing edge retraction (Figure 5b, S31b-e). Eventually, the suppression effect on cell motility was remained (Figure 5c, S31f-h). To preserve FAs on cell edge with actomyosin-mediated force transmission, we applied Rho Activator II on HuH-7 cells to drive elevated level of actomyosin contractility.(27, 28)

Revised Figure and caption:

Figure 5. Overcome the influence of super high density of ligands via Rac1 signaling. (a) F-actin phalloidin (magenta), vinculin N-terminal domain lacking the tail domain (vin258) (cyan), and paxillin (yellow) immunofluorescence in HuH-7 cell expressing pEGFPC1/GgVcL 1-258 upon the treatment of FFFIKLLI (100 μM) for 12 hr. Scale bar represents 5 μm . (b) Kymograph of normalized edge velocity and mean velocity over time for protrusions and retractions of HuH-7 cells expressing pEGFPC1/GgVcL 1-258 with or without the treatment of FFFIKLLI for 12 hr. (c) The trajectory plots of ~ 200 randomly selected migrating HuH-7 cells expressing pEGFPC1/GgVcL 1-258 with or without the treatment of FFFIKLLI. Scale bars represent 50 μm . F-actin phalloidin staining (magenta) and paxillin (green) immunofluorescence (d), kymograph

of normalized edge velocity and mean velocity over time for protrusions and retractions (e), and the trajectory plots (n = ~200) (f) of HuH-7 cells pretreated with Rho Activator II (1 µg/mL) with or without 12 hr treatment of FFFIKLLI (100 µM). F-actin phalloidin staining (magenta) and paxillin (green) immunofluorescence (g), kymograph of normalized edge velocity and mean velocity over time for protrusions and retractions (h), and the trajectory plots (n = ~200) (i) of HuH-7 cells with elevated Rac1 activation with or without 12 hr treatment of FFFIKLLI (100 µM). (j) Schematic summary of outside-in regulation of HuH-7 cell motility via precise control of ligand density on nanofilaments, and the restoration of cell motility via intracellular signaling.

[12. The authors claimed that Rho activator II can rescue the effect of FFFIKLLI in membrane retraction, and Tiam1-induced Rac1 activation can restore the effect of FFFIKLLI in the membrane protrusion. Does FFFIKLLI inhibit the activity of RhoA and Rac1?]

The FFFIKLLI self-assembled nanofilaments suppressed Rac1 activity that was confirmed via live-cell imaging of a FRET-based Rac1 biosensor. The experimental results were presented in Figure 4d, 4e by following the published protocol (*Mol. Biol. Cell* 2011, 22, 4647). The correlated information was presented in the Supporting Information. We also conducted the experiments to examine the effect of FFFIKLLI on RhoA activity (Figure S30b, S30d), and the results indicated that FFFIKLLI suppressed RhoA activity.

Addition in the Supporting Information:

Fig. S30.

Representative FRET/CFP ratio images and quantitative analysis of HuH-7 cells expressing RaichuEV-Rac1 or DORA-RhoA with or without the treatment of peptide assemblies for 12 hrs. The images were coded according to a pseudo color scale, which ranges from yellow to purple with an increase in FRET activity. Scale bars represent 20 μ m. $n=50$ cells for each group. Kruskal-Wallis with Dunn's multiple comparisons test was used for analysis of the data. Error bars represent s.d..

Original:

Together with a great inhibition of Rac1 activity (Figure 4d) on cell periphery (Figure 4e) which indicated the prevention of both protrusion formation and forward motion, it was demonstrated that self-assembly of FFFIKLLI restricted both trailing edge retraction and leading-edge protrusion of HuH-7 cells resulting into depolarization suppressing cell motility.

Revised:

Together with a great inhibition of Rac1 and RhoA activity on cell periphery (Figure 4d, 4e, and S30) which indicated the prevention of both protrusion formation and forward motion, it was demonstrated that self-assembly of FFFIKLLI restricted both trailing edge retraction and leading-edge protrusion of HuH-7 cells resulting into depolarization suppressing cell motility.

[Does FFFIKLLI regulate different signaling pathways in different regions of a cell?]

The FFFIKLLI self-assembled nanofilaments cause global inhibition of RhoA and Rac1 activity.

[Do the authors quantify membrane protrusion and retraction in the leading edge or trailing edge in Figure 4e and Figure 4h? The authors should show representative images and videos.]

In Figure 4e and Figure 4h (Figure 5e and Figure 5h in revised manuscript), we performed morphodynamical analysis by mapping the protrusion and retraction along the entire periphery of the cell. The correlated velocity analysis demonstrated the membrane dynamics of the entire periphery.

The representative time-lapse images, correlated experimental results and data analysis were summarized in Figure S32 and S33 in the Supporting Information. We also revised the main text for a clearer indication of the experimental data.

Original:

Followed by 12 hr treatment of nanofilaments, the FAs remained on cell periphery associated with peripheral actin bundles (Figure 4d, S21a-d). Enhanced contractile forces eased the full disassembly of FAs facilitating trailing edge retraction, which was confirmed by the velocity profile of edge dynamics (Figure 4e). However, the protrusion activity was still restricted (Figure 4e) resulting into partial restoration of cell motility (Figure 4f, S21e-g). Because Rac1 can trigger new leading-edge formation by controlling local actin assembly and FA formation when activated at the cell edge,(29) we then activated Rac1 constantly by translocating Tiam1 to the plasma membrane of HuH-7 cells to compel the formation of lamellipodia (Figure S22). Upon the treatment of nanofilaments for 12 hr, cells exhibited both NAs localized across the lamellipodia, and FAs associated with F-actin bundles collected in a transverse band (Figure 4g, S23a). Both leading-edge protrusion and trailing edge retraction were not restricted (Figure 4h, S23b-f), same as the cell motility (Figure 4i).

Revised:

Followed by 12 hr treatment of nanofilaments, the FAs remained on cell periphery associated with peripheral actin bundles (Figure 5d). Enhanced contractile forces eased the full disassembly of FAs (Figure S32a, b) facilitating trailing edge retraction, which was indicated by the time-lapse images (Figure S32a) and confirmed by the velocity profile of edge dynamics (Figure 5e). However, the protrusion activity was still restricted resulting into partial restoration of cell motility (Figure 5f, S32c-h). Because Rac1 can trigger new leading-edge formation by controlling local actin assembly and FA formation when activated at the cell edge,(29) we then activated Rac1 constantly by translocating Tiam1 to the plasma membrane of HuH-7 cells to compel the formation of lamellipodia (Figure S33a). Upon the treatment of nanofilaments for 12 hr, cells exhibited both NAs localized across the lamellipodia, and FAs associated with F-actin bundles collected in a transverse band (Figure 5g). Both leading-edge protrusion and trailing edge retraction were not restricted (Figure 5h, S33b, S34a-c), same as the cell motility (Figure 5i, S34d-g).

[13. The authors claimed that constant activation of Rac1/Tiam1 signaling was the only effective

rescue from in-side out. In-side out signals mean activation of integrin from intracellular domain of integrin. So, how does Rac1/Tiam1 signals activate integrin in-side out? Does talin or kindlin involve? What is the role of integrin $\alpha 3 \beta 1$ in the cells with Tiam1-mediated Rac1 activation? Does Tiam1-mediated Rac1 activation can regulate FFFIKLLI-mediated integrin signals?]

We are sorry for causing misunderstanding and confusion by summarizing the results imprecisely. First, we should not use 'only' here. Besides that, we should not use 'from in-side out' which caused confusion with 'in-side out activation of integrin'. The fact is we tried three methods to rescue the cell motility intracellularly. And among these three methods, only constant activation of Rac1/Tiam1 signaling can fully restore the cell motility.

In this research, our focus is to demonstrate the excessive binding interactions between integrin and nanofilaments with super high ligand density which is applicable as a potential strategy to suppress cancer metastasis. The outside-in activation of integrin via ligand binding was utilized in the research, for instance, by applying FFFIKLLI assembled nanofilaments in HuH-7 cell culture, the integrins on the cells were activated via ligand binding extracellularly. And because the integrins of suppressed cell had been activated, we didn't consider 'in-side out activation of integrins' as one of the options to rescue the cell restoring its motility. Overall, in-side out activation of integrin was not involved and it's beyond the scope of this manuscript. Based on the comments of the reviewer, we revised the main text to summarize the results accurately.

Original:

To understand the influence of super high ligand density on a molecular level, we did further exploration on the inside-out signaling to overcome the outside-in restriction.

Revised:

To understand the influence of super high ligand density on a molecular level, we did further exploration on the activation of intracellular signaling to overcome the outside-in restriction.

Original:

Together, the excessive binding interactions between integrins and the super high-density ligands deactivated the endogenous Rac1 effectively via an outside-in path. Besides reducing ligand density on nanofilaments extracellularly, constant activation of Rac1/Tiam1 signaling was the only effective rescue from in-side out. In regard to Rac1's particular roles for tumor metastasis, the therapeutic potentials of super high ligand density are promising.

Revised:

Together, the excessive binding interactions between integrins and the super high-density ligands globally deactivated the endogenous Rac1 via an outside-in path. Besides removing the nanofilaments or lowering the ligand density on nanofilaments extracellularly, constant activation of Rac1/Tiam1 signaling was an effective intracellular rescue. Regarding Rac1's particular roles for tumor metastasis, the therapeutic potentials of super high ligand density are promising.

(Minor comments)

[1. Figure S1g-i is not shown in Figure S1.]

Figure S1g-i has been rearranged as Figure S3a-c. We corrected the figure number in the main text.

[2. Line 71, Figure S4 should be Figure S4 and S5.]

Thank you for the comment. We corrected it in the main text.

[3. For the experiments of wound healing, the authors should show the results as "percentage of wound closure", include representative images, and have statistical analysis.]

Following the reviewer's comments and suggestions, we revised the figure presentation and included the representative images, statistical analysis in the Supporting Information.

Revised Figures:

Fig. S2.

(a) HuH-7 cell viability using MTT assay. Cells were incubated with 10, 20, 50, 100, 200 μM of FFFIKLLI for 24, 48, or 72 hr. Error bars represent s.d.. (b-d) Wound closure rate and representative images of HuH-7 cells with or without the treatment of FFFIKLLI at 10, 20, 50, 100, and 200 μM . Error bars represent s.d.. (e-g) Wound closure rate and representative images of HuH-7 cells with or without the treatment of FFF, IKLLI, FFFIKLLI, and FFFKLIL at 100 μM . Error bars represent s.d..

Fig. S3.

Wound closure rate and the representative images of peptide FFFIKLLI treatment on HeLa, Hep G2, A549, MKN1, U-87 MG, MCF-7 and Ect1/E6E7 cell lines. Error bars represent s.d..

Fig. S8.

Representative images and relative wound closure rate of HuH-7 cells upon the treatment of co-assembled FFFIKLLI (fixed at 100 μ M) and FFF at various ratios.

[4. To show the expression of integrin $\alpha 3 \beta 1$ in different cell lines, RT-PCR is required.]

To address the reviewer's suggestion, we added the RT-PCR results in Figure S4, and the assay in the Supporting Information.

Fig. S35 The other assembling ligands affect cell migration.

(a-b) Representative images and 48hr-Wound healing rate of HuH-7 cells upon the treatment of 200 μ M assembling ligands. **(c)** The phalloidin staining of HuH-7 cells incubated with 200 μ M assembling ligands for 12 hr. Scale bar represents 20 μ m. **(d)** Immunofluorescence of paxillin co-stain with phalloidin. HuH-7 cells were incubated with 200 μ M assembling ligands for 12 hr. Scale bar represents 20 μ m.

Addition in the Supporting Information:

Quantitative RT-PCR

Total RNA was immediately extracted from cultured cells using TRIzol reagent (Invitrogen). cDNA was synthesized with the cDNA Synthesis Kit (Bio-Rad). qRT-PCR was performed on a qTOWER 3 Real-Time Thermal Cyclers (Analytik Jena) using Power SYBR Green Master Mix (ThermoFisher Scientific). Gene expression of integrin was normalized using the comparative Ct quantification method. And the primer sequences was used as follows:

hGAPDH: 5'-GGCATCCTGGGCTACTGA-3'(F), 5'-GAGTGGGTGTCGCTGTTGAA-3'(R);
hITGA3:5'-GCCTGACAACAAGTGTGAGAGC-3' (F), 5'-GGTGTTTCGTCACGTTGA TGCTC-3' (R);
hITGB1: 5'-GGATTCTCCAGAAGGTGGTTTCG-3' (F), 5'-TGCCACCAAGT TTCCCATCTCC-3' (R).

Fig. S4.

(a-b): mRNA expression of integrin β_1 and integrin α_3 in different cell lines by quantitative RT-PCR analysis. Kruskal-Wallis with Dunn's multiple comparisons test was used for analysis of the data. Error bars represent s.d.. (c): Protein expression of integrin β_1 and integrin α_3 in different cell lines by western blotting analysis. (d): Cell surface protein expression of integrin β_1 and integrin α_3 in HuH-7, HeLa, MCF-7 and Ect1/E6E7 cell lines by flow cytometry analysis. (e-f): Representative images and quantitative analysis showing attached HuH-7 cells in plates coated with or without 100 μ M FFFIKLLI. Scale bar represents 100 μ m. n = 30 for each group. Mann-Whitney test was used for analysis of the data. Error bars represent s.d.. (g-i): Knockdown efficiency of integrins in HuH-7. (j, k): 48 hr wound healing rate of peptide FFFIKLLI treatment on integrin knockdown HuH-7 cells. Peptide concentration, 100 μ M. Scale bar represents 100 μ m. Kruskal-Wallis with Dunn's multiple comparisons test was used for analysis of the data. Error bars represent s.d.. (l, m): Representative images and quantitative analysis showing attached HuH-7 cells in plates coated with 100 μ M FFFIKLLI. Scale bar represents 100 μ m. n = 30 for each group. Kruskal-Wallis with Dunn's multiple comparisons test was used for analysis of the data. Error bars represent s.d..

[5.Line 174, please check if pEGFPC1-mEGFP-paxillin is correct.]

The pEGFPC1-mEGFP-paxillin expression vector was a gift from A. Kusumi, and was reported on *Nature Chemical Biology* (2018, 14, 497-506). As reported, 'the sequence encoding mGFP-paxillin was subcloned into the pEGFP vector (Clontech)'. And this is the reason why we wrote in this way. However, the authors used mGFP-paxillin as the name of the vector. To let readers easily find the related details, we used mGFP-paxillin to replace pEGFPC1-mEGFP-paxillin and cited the reference paper.

[6.Line222, NAs should be FAs.]

We corrected it to FAs.

[7.For the TIAM1 membrane translocation, the authors should include the details in the materials and methods.]

The experimental details were added in the Supporting Information as below:

Addition in the Supporting Information:

For the Rac1 activation, HuH-7 cells were transfected with Lyn11-linker-FRB, YFP-FKBP-linker-Tiam and mRuby-Lifeact-7 in a 2:1:1 ratio. One or two days after transfection, the cells are set to a time-lapse microscope, and treated with 100 nM Rapamycin. And the Tiam1 was translocated to the membrane within 2 mins to locally activated Rac1 signaling.

Reviewer #2 (Remarks to the Author):

[This article presents a study about mechanistic understanding of cell migration through engineering integrin ligand assembly. The authors propose a bottom-up fabrication strategy to enhance the resolution of such systems to the molecular level. The manuscript is coherent and well-written. However, there are questions that the reviewer feels need to be investigated and answered before the article's significance can be established:]

We appreciate the reviewer's suggestions. Following that, we carefully addressed the raised questions point-by-point as shown below.

[-The Introduction part of the paper needs more references and clarification on previous work and how this work is different from/improving upon the past work.]

We thank the reviewer's comment. Following the suggestion, we revised the introduction to address the importance and novelty of this research in comparison to the previous works.

Original:

Cell migration plays a central role in a wide variety of biological phenomena, from embryogenesis to tumor metastasis, etc.(1) There is considerable interest in understanding cell migration on a molecular level because this could lead to novel therapeutic approaches in biotechnology.(2) Integrins, as the major family of cell receptors responsible for cell adhesion, have long served as the primary targets of biomaterials.(3-5) The development of top-down nanofabrication techniques(6, 7) enhanced control over spatial presentation of integrin ligand, which promoted the mechanistic study of integrin-mediated adhesions to inspire biomaterial innovations. For example, fibronectin (50 µg/ml)-coated polystyrene microbeads (mean diameter 11.9 µm) facilitated the elucidation of synergic effects of integrin occupancy and aggregation on cellular response,(8) and RGD-functionalized Ti lithography nanopattern (10 nm lines with 40-490 nm distance) assisted the demonstration of ligand geometrical effects on adhesion cluster formation.(9) Enhancing spatial resolution beyond sub-micron for insights of subsequent cellular response will reveal design principles for future generations of biomaterials.

To address the challenges, we develop a bottom-up fabrication strategy⁽¹⁰⁾ by combining molecular self-assembly and co-assembly⁽¹¹⁾ for extracellular constructs.

Revised:

Cell migration plays a central role in a wide variety of biological phenomena, from embryogenesis to tumor metastasis, etc.⁽¹⁾ There is considerable interest in understanding cell migration on a molecular level because this could lead to novel therapeutic approaches in biotechnology.⁽²⁾ Integrins, as the major family of cell receptors responsible for cell adhesion and migration, have long served as the primary targets of biomaterials.⁽³⁻⁵⁾ Initially, the modulation of the adhesion surface relies on the control of the global density of integrin ligands.⁽⁶⁾ Following the development of polymer blending technique, the first generation of materials displaying multivalent ligands were synthesized, which signified the necessity of the regulation of ligand local density.^(7, 8) Two decades ago, fibronectin (50 µg/ml)-coated polystyrene microbeads were fabricated facilitating the elucidation of synergic effects of integrin occupancy and aggregation on cellular response.^(9, 10) About ten years ago, RGD-bound gold nanoparticles were fabricated as anchor points for single integrin $\alpha_v\beta_3$. Combined with block-copolymer micelle nanolithography, patterned surfaces with variable global ligand density ranging from 52 to 367 μm^{-2} and variable ligand spacing ranging from 50 to 135 nm were produced revealing the crucial influence of ligand spacing on $\alpha_v\beta_3$ integrin-mediated cell adhesion. Recently, via nanoimprint lithography, the RGD-bound 10-nm lines functioned as linear arrays of single integrin $\alpha_v\beta_3$ binding sites in single, crossing, or paired patterns with 40-490 nm distance were produced assisting the demonstration of ligand geometrical effects on adhesion cluster formation.⁽¹¹⁾

The development of nanofabrication techniques^(12, 13) enhanced control over spatial presentation of integrin ligands, which promoted the mechanistic study of integrin-mediated adhesions to inspire biomaterials innovations. For instance, besides the global ligand density, the local ligand density, the size of 'ligand island', the spacing between islands, and the ligand spacing all became critical parameters in materials design to control cell adhesion. Undoubtedly, enhancing the control of ligand presentation beyond the current resolution of top-down nanofabrication for insights of subsequent cellular response will reveal design principles for future generations of biomaterials. To address the challenges, we develop a bottom-up fabrication strategy reaching molecular level resolution ⁽¹⁴⁾ by combining molecular self-assembly and co-assembly⁽¹⁵⁾ for extracellular constructs. Compared to the top-down techniques, the proposed bottom-up strategy does not require high-cost equipment nor rigorous fabrication condition. Essentially, the simple formular of molecular assembly, as a practical and readily applicable approach, eases the boundary between fundamental study and biomedical applications.

[-The authors should provide a more in-depth physical insight about how raising the proportion of FFF may lead to the suppression effect on cell migration.]

We thank the reviewer's comments. Before addressing the questions point-by-point, we would like to clarify that raising the proportion of FFF attenuates the suppression effect on cell migration. And raising the proportion of FFF to a certain high level can even promote cell

migration. Instead, high proportion of FFFIKLLI leads to the suppression effect on cell migration.

There is the consensus that via the control of ECM ligand density, cell migration can be regulated, which has been verified by biologists through varying the weight/volume of fibronectin applied for substrate coating (Cell, 2006, 125, 1361-1374). Consistent to that, changing the proportion of FFF alters the density of integrin ligand (IKLLI) on the co-assembled nanofilaments, which leads to various influences on cell migration.

[For example, what is special about 1 to 44, 1 to 89 or 1 to 249 ratio values that they discuss.]

At the ratio of 1 to 249, the total concentration of two components is 2.5 mM, which is close to saturation condition for the fabrication of assembled nanofilaments. Therefore, we expanded the variation range to the ratio of 1 to 249. Within the range, we select the ratio values including 1 to 0 (1), 1 to 1 (1/2), 1 to 2 (1/3), 1 to 4 (1/5), 1 to 1 to 9 (1/10), 1 to 19 (1/20), 1 to 44 (1/45), 1 to 89 (1/90), 1 to 149 (1/150), and 1 to 179 (1/180), and 0 to 249 as pure FFF condition to cover the whole range for varieties (the numbers in the squares represent the molar ratio of FFFIKLLI to the total of the two components). As summarized in revised Figure S8, the combination of wound healing experimental results upon these treatment conditions indicated that adjusting the proportions of the two components could create nanofilaments that lead to suppression, neutral, to promotion effect on cell migration. Although the ratio values, 1 to 44, 1 to 89, and 1 to 249, don't serve as the critical ratio values, approximately from 1 to 44 to 1 to 249 indicate a range leading to promotion effect on cell migration.

[Have they performed any simulations or analytical modeling to consider various ranges of the ratios and observe how changing the ratio affects the system dynamics?]

The single crystal structure of FFF in self-assembled nanofilament was determined by single crystal XRD measurements. Based on that, we did molecular dynamics simulation and polymorph prediction for FFFIKLLI self-assembly and FFFIKLLI/FFF co-assemblies at 1 to 1, 1 to 2, and 1 to 4 ratios using Materials Studio 2020. For the ratio value of 1 to 44 to 1 to 249, we did statistical estimation of the surface ligand density by using the crystal unit cell of FFF for molecular packing. Please refer to the details in the response to the revised 'Molecular dynamics simulation and polymorph prediction' and the correlated figures and table in the supporting information. The estimation results indicated that increasing the proportion of FFF, the surface density of integrin ligand (IKLLI) is induced and the distance between ligands is increased. Therefore, changing the proportion of the two components affects the spatial distributions of integrin ligands leading to different effects on cell migration, which is consistent to the published results revealed by biologists (Cell, 2006, 125, 1361-1374).

[How are the estimated molecular packing structures obtained for Figure 1b sketches?

-Along the same lines, for quantitative estimation of surface ligands, what kind of initial configuration/setup was used within MD simulations? Is this what Figure S8 is trying to explain? Were the simulations fully atomistic or coarse-grained? Was there a timestep used within the MD simulations or were all simulations performed in molecular statics form for energy minimization purposes? What procedure/measure was used to calculate surface ligand density from the simulation results? Unless there are page/figure-number limitations specified by the journal, I

would recommend the authors produce a separate Figure describing the procedure and the results of the simulations.]

We thank the reviewer's comments. For the quantitative estimation of surface ligands, the initial configuration/set up was based on the reported FFF conformation and the single crystal unit cell of FFF after optimization via Focite module as described in the 'Molecular dynamics simulation and polymorph prediction' section of supporting information which is also revised as below by following the reviewer's suggestion. Figure S8 represents the initial configuration of FFFIKLLI, FFFIKLLI/FFF at 1 to 1, 1 to 2, 1 to 4 ratios, and FFF for Polymorph Prediction using Materials Studio. The simulation is fully coarse-grained. And all simulations were performed in molecular statics form for energy minimization.

The previous estimation of ligand distribution on co-assembled nanofilament formed from FFFIKLLI/FFF at 1 to 249 ratio was calculated based on a 3D packing of FFF unit cells aligned to a fixed height of 10 nm (the minimum height of co-assembled nanofilament obtained from AFM tests). Since various ratio values lead to nanofilaments with different heights, we changed the statistical estimation to single layer of crystal unit cells for comparison among the ratio values from 1 to 44 to 1 to 249 as presented in Figure S15.

Following the reviewer's suggestions, we revised the Figure 2b, Figure S11, and the 'Molecular dynamics simulation and polymorph prediction' section. We also added step-by-step results for the procedure description.

Revised Figure 1b:

Figure 1. Engineering integrin ligand assembly to control ligand presentation. (a) Schematic illustration of precise control of integrin ligand presentation on nanofilaments via peptide assembly. (b) TEM images of nanofilaments obtained via molecular self-assembly and co-assembly of FFFIKLLI (100 μM) and FFF at various ratios, and the estimated molecular packing structures. IKLLI motif is presented in blue and FFF motif is presented in pink. The scale bars represent 200 nm. (c) Zoom-in SEM images (false color) of HuH-7 cell edge and apical membrane after 3-day incubations. FFFIKLLI was maintained at a concentration of 100 μM . Cell body is highlighted in pink, while the nanofilaments are highlighted in blue. The scale bars represent 300 nm.

Revised Figure S14:

Fig. S14.

Space-filling model of surface structure formed by extending the unit cells of polymorph predictions. FFF motif was presented in pink, while IKLLI motif was presented in blue.

Original:

Molecular dynamics simulation and polymorph prediction

Molecular mechanics calculations were performed using Materials Studio. For all simulations, the Ewald method (3,4) was used for the electrostatic and van der Waals interaction terms. Gasteiger charges were used for an initial conformational search. As the crystal structure prediction method uses a rigid body approximation in the initial search for crystal packing alternatives, the analysis to determine low energy geometry was performed by following the protocols reported by Kim etc. (5), and the results were used as input for the packing calculations. The conformation of FFF was reported by Ellenbogen etc(6). Therefore, FFFIKLLI was drawn based on the structure of the FFF motif, and geometrical energy minimization scans were performed using the Forcite module of Materials Studio. After finding the lowest energy conformation of FFFIKLLI, the reported structure of the FFF unit cell was used as the starting

point for crystal structure prediction using the Materials Studio Polymorph Predictor (PP). By replacing FFF by FFFIKLLI from the reported unit cell, PP calculation was performed.

The PP was set to its default fine setting (this sets the simulated annealing algorithm to a temperature range of 300-100000.0 K with a heating factor of 0.025, requiring 12 consecutive steps to be accepted before cooling and a maximum of 7000 steps) with the force field Dreiding 2.21 with Gasteiger charges. The 10 most common space groups found in organic crystals registered in the CSD were selected, including $P2_1/c$, $P1$, $P2_12_12_1$, $P2_1$, $C2/c$, $Pbca$, $Pna2_1$, $Pbcn$, Cc , and $C2$. Clustering of the predicted polymorphs was done using the polymorph clustering routine in Materials Studio. After the final clustering, hydrogen bonding analysis was performed on the calculated crystal structures to identify the packing modes matching the FTIR spectra regarding the hydrogen bonding signals. After extending the structure along the unit axes, the surface that exposes most integrin ligand IKLLI was presented in a defined square area aligned to the self-assembled nanostructures. The molecular packing of the mixture of FFFIKLLI and FFF at 1:249 ratio was predicted based on the crystal structure of FFF unit cell and the molecular packing structure of 1:4 ratio. The ligand distance was measured using the distance measurement functions of Materials Studio.

Revised:

Molecular dynamics simulation and polymorph prediction

Molecular mechanics calculations were performed using Materials Studio[®] 2020. For all simulations, the Ewald method (3,4) was used for the electrostatic and van der Waals interaction terms. Gasteiger charges were used for an initial conformational search. As the crystal structure prediction method uses a rigid body approximation in the initial search for crystal packing alternatives, the analysis to determine low energy geometry was performed by following the protocols reported by Kim etc. (5), and the results were used as input for the packing calculations.

The conformation of FFF was reported by Ellenbogen etc(6). As reported, the single crystal structure of FFF in FFF-tape (self-assembled nanostructure) was determined by single crystal XRD measurements to 1.1 Å resolution. The determined structure is triclinic, space group $P1$, with four FFF molecules per asymmetric unit. The alignment of the unit cell with regards to the self-assembled tape structure reveals that growth is governed by π -interactions between adjacent aromatic rings along the c-axis.

1. Import the crystal structure of FFF to Materials Studio and draw FFFIKLLI based on the conformation of FFF. The geometrical energy minimization scans were performed using the Forcite module of Materials Studio. The molecule in Fig. S11a was found to have the lowest energy conformation. The force field used was Dreiding 2.21 with Gasteiger charges as implemented in the Materials Studio packages.
2. Import the FFF unit cell obtained via single crystal XRD measurements as described in the protocol to Materials Studio. Replacing FFF by optimized FFFIKLLI to 1:1, 2:1 and 4:1 ratio, respectively. The initial placement of FFFIKLLI was determined by π -interactions between adjacent aromatic rings along the c-axis. The geometrical energy minimization scans were performed using the Forcite module. The optimized gas phase conformations as presented in Fig. S11b were used as the starting points for crystal structure prediction using the Materials Studio Polymorph Predictor (PP).
3. The PP was set to its default fine setting (this sets the simulated annealing algorithm to a temperature range of 300-100000.0 K with a heating factor of 0.025, requiring 12

consecutive steps to be accepted before cooling and a maximum of 7000 steps) with the force field Dreiding 2.21 with Gasteiger charges. The 10 most common space groups found in organic crystals registered in the CSD were selected, including *P2₁/c*, *P1*, *P2₁2₁2₁*, *P2₁*, *C2/c*, *Pbca*, *Pna2₁*, *Pbcn*, *Cc*, and *C2*. Clustering of the predicted polymorphs was done using the polymorph clustering routine in Materials Studio. After the final clustering, hydrogen bonding analysis (as implemented in the Materials Studio packages) was performed on the calculated crystal structures (Fig. S12). According to the FTIR spectra, self-assembled FFF presents NH-O hydrogen bonding, self-assembled FFFIKLLI presents NH-N hydrogen bonding, co-assembled FFF and FFFIKLLI present both NH-O and NH-N hydrogen bonding. The reported FFF unit cell shows both intermolecular and intramolecular NH-O hydrogen bonding which matches to the FTIR results. Based on the summarized hydrogen bonding analysis of the calculated crystal structures (Table S1), we highlighted the structures that match to the FTIR results in black frames.

4. The TEM images and SEM images of self-assembled FFF, self-assembled FFFIKLLI, co-assembled FFF with FFFIKLLI demonstrated that the assembled nanofilaments all shared similar morphologies. Taking advantage of unified space group symmetry to reduce the number of variables in searches of molecular packing modes in nanofilaments assembled by FFF and FFFIKLLI at various proportions, we selected the clustering results of *Pbcn* space group (Fig. S13) which generated matching structures at different proportions to ease the comparisons in regard of ligand (IKLLI) distribution density.
5. After extending the structure along the unit axes, the surface that exposes most integrin ligand IKLLI was presented in a defined square area (10 x10 nm²) within the dimension range of nanofilaments (Fig. S14). The ligand density on the surface of nanofilaments formed by co-assembly of FFFIKLLI and FFF at 1:44, 1: 89, 1 to 149, 1 to 179, and 1:249 ratio was estimated statistically based on the crystal structure of FFF unit cell by replacing one FFF with FFFIKLLI on the filament surface that composed of 45, 90, 150, 180 and 250 FFF, respectively. The detailed calculation results are presented in Fig. S15.
6. The exposed ligand IKLLI with terminal group protruding out of the surface area was identified as effective ligand, and the distance between the effective ligands was measured using the distance measurement functions of Materials Studio by calculating the distance between the C-terminals of the effective ligands (Fig. S14).

Addition in the Supporting Information:

Figure S11. The molecular and geometry-optimized structures of FFFIKLLI (a), FFF and FFFIKLLI at 1:1, 1:2, and 1:4 ratio (b) presented in stick model.

d

Fig. S12.

Predicted crystal structure unit cell of FFFIKLLI (a), FFFIKLLI:FFF = 1:1 (b), FFFIKLLI:FFF = 1:2 (c), FFFIKLLI:FFF = 1:4 (d), with the $P2_1/c$, $P1$, $P2_12_12_1$, $P2_1$, $C2/c$, $Pbca$, $Pna2_1$, $Pbcn$, Cc , and $C2$ space group symmetry. The structures are presented in stick model.

Table S1. Hydrogen bonding analysis of predicted crystal structure unit cells. Intra represent intramolecular hydrogen bonding and inter represent intermolecular hydrogen bonding.

Space group	FFF		FFFIKLLI:FFF = 1:4		FFFIKLLI:FFF = 1:2		FFFIKLLI:FFF = 1:1		FFFIKLLI	
	NH-O	NH-N	NH-O	NH-N	NH-O	NH-N	NH-O	NH-N	NH-O	NH-N
XRD intra	✓									
XRD inter	✓									
C2-C intra			✓		✓	✓	✓	✓	✓	
C2-C inter					✓					
C2 intra			✓		✓	✓	✓	✓	✓	
C2 inter			✓						✓	
CC intra			✓		✓	✓	✓	✓	✓	
CC inter					✓					
P-1 intra			✓		✓		✓		✓	
P-1 inter				✓			✓			
P21-C intra			✓		✓	✓	✓			
P21-C inter			✓	✓	✓					
P21 intra			✓		✓	✓	✓		✓	✓
P21 inter			✓	✓	✓	✓	✓			
P212121 intra			✓	✓	✓		✓		✓	✓
P212121 inter			✓	✓	✓				✓	
PBCA intra			✓		✓	✓	✓			✓
PBCA inter			✓		✓	✓	✓		✓	
PBCN intra			✓		✓	✓	✓	✓		✓
PBCN inter				✓	✓		✓			
PNA21 intra			✓		✓		✓		✓	✓
PNA21 inter						✓	✓		✓	✓

Fig. S15. (a) Space-filling model of crystal structure unit cell of FFF at zy, zx, xy plain. FFF motif was presented in pink. (b) Because the C-terminus of FFF that can be covalently linked with IKLLI is only exposed toward the zx plain (the image in the middle), we took the area size of this plain for surface calculation. The calculation results of ligand density for FFFIKLLI/FFF at 1 to 249 to 1 to 44 were summarized. (c) The scheme represents the estimation of ligand distance in regard of three different densities.

[- As a minor point, in Figure 1b,d,e, the text is very small and almost not readable.]

Following the reviewer's comments, we modified Figure 1 by dividing it to two figures imbedded with text of larger font size.

Revised:

Figure 1. Engineering integrin ligand assembly to control ligand presentation. (a) Schematic illustration of precise control of integrin ligand presentation on nanofilaments via peptide assembly. (b) TEM images of nanofilaments obtained via molecular self-assembly and co-assembly of FFFIKLLI (100 μ M) and FFF at various ratios, and the estimated molecular packing structures. IKLLI motif is presented in blue and FFF motif is presented in pink. The scale bars represent 200 nm. (c) Zoom-in SEM images (false color) of HuH-7 cell edge and apical membrane after 3-day incubations. FFFIKLLI was maintained at a concentration of 100 μ M. Cell body is highlighted in pink, while the nanofilaments are highlighted in blue. The scale bars represent 300 nm.

Figure 2. Nanofilaments with various ligand presentations regulate both cell shape and cell migration. (a) The phalloidin staining of Huh-7 cells with and without the treatment of various nanofilaments. Scale bars represent $20\ \mu\text{m}$. (b) The spreading area and the perimeter area ratio of HuH-7 cells under various conditions. Kruskal-Wallis with Dunn's multiple comparisons test was used for analysis of the data. Error bars represent standard deviation. From left to right, $n = 61, 49, 45, 46, 51, 56$ cells, respectively. The trajectory plots (c), and the correlated quantitative analysis of travel speed, persistence and persistence index (d) of randomly selected migrating cells for each incubation condition. Live cell images were taken every 10 min for a total of 10 hr. Kruskal-Wallis with Dunn's multiple comparisons test was applied in

data analysis. Error bars represent standard error of mean. From left to right, $n = 261, 280, 230, 260, 214,$ and 278 cells, respectively. Scale bars in panel c represent $50 \mu\text{m}$.

[- I find the discussion about the results of Figure 2 somewhat lacking. What do the authors mean with "super high" and "low" ligand densities? Can they clarify/quantify this parameter based on their measurements or any simulations?]

We thank the reviewer's constructive suggestion. Following that, we revised the main text by presenting the reported research results on ligand densities achieved by far in the introduction part (please refer to the revised introduction presented as response to the first question). As reported, the highest global ligand density was $52 \text{ per } \mu\text{m}^2$, and shortest ligand distance was 10 nm . Compared to that, the self-assembly of FFFIKLLI displayed the highest ligand density by far. Scientists reported three phenotype FA organizations in response to high, intermediate, and low ECM (especially fibronectin) concentration (Cell, 2006, 125, 1361-1374). Following that, the ligand density of co-assemblies that induced the same FA organization is categorized into high, intermediate, and low level, respectively. And the self-assembled FFFIKLLI with the highest ligand density record inducing non-reported unique FA organization was defined as super high ligand density. Overall, by comparing to the previous studies, based on the simulation results and the correlated influence on cell migration, we defined the ligand density level in this research.

To address the reviewer's question, we revised the paragraph for the discussion of Figure 2.

Original:

Quantitative estimation of surface ligand density of nanofilaments was conducted via molecular dynamics simulation based on the crystal structure of FFF unit cell,⁽¹⁶⁾ followed by polymorph prediction. After the initial search, Fourier-transform infrared (FTIR) spectra of nanofilaments, which indicated the hydrogen bonding transition from N-H...N to N-H...O (20) due to the increasing proportion of FFF (Figure S7) were applied to select the adaptive packing modes (Figure S8). The polymorph predictions suggested that the molecular packing of self-assembled FFFIKLLI could expose $48 \text{ ligands}/100 \text{ nm}^2$, which is the highest record of ligand density, with the shortest distance between ligands ranging from 0.5 to 1.6 nm . Co-assembly of FFFIKLLI with FFF leads to decreased ligand density with increased ligand distance. For example, from 1 to 1 , to 1 to 4 , and to 1 to 249 ratio, the estimated ligand density decreased from 20 to 12 to $3 \text{ ligands per } 100 \text{ nm}^2$ with minimum ligand distance increased from 1.1 to 2.2 to 4.9 nm , respectively (Figure 1b, S9). Considering their influence on cell migration, we here categorize the ligand presentation on nanofilaments into four levels. Self-assembled FFFIKLLI possesses super high ligand density; co-assembled FFFIKLLI and FFF at 1 to 1 and 1 to 2 ratios possesses high ligand density; co-assembly at 1 to 4 ratio possesses intermediate ligand density; and co-assembly at 1 to 249 ratio possesses low ligand density.

Revised:

Quantitative estimation of ligand density on the surface of nanofilaments was conducted using molecular dynamics simulation and polymorph prediction. Based on the crystal structure of FFF under self-assembling condition (16), the crystal structure prediction using a rigid body approximation was applied in the initial search for crystal packing alternatives. Fourier-

transform infrared (FTIR) spectra of nanofilaments, which indicated the hydrogen bonding transition from N-H...N to N-H...O (20) due to the increasing proportion of FFF in molecular assemblies (Figure S7), were applied to select the adaptive packing modes (Figure S10). The polymorph predictions suggested that the molecular packing of self-assembled FFFIKLLI could possibly expose 48 ligands per 100 nm² with ligand spacing ranging from 0.5 to 1.6 nm, which is the highest record of ligand density, with the shortest distance between ligands ranging from 0.5 to 1.6 nm. Co-assembly of FFFIKLLI with FFF leads to decreased ligand density with increased ligand distance. For example, from 1 to 1, to 1 to 4, and to 1 to 249 ratio, the estimated ligand density decreased from 20 to 12 to 3 ligands per 100 nm² with minimum ligand distance increased from 1.1 to 2.2 to 4.9 nm, respectively (Figure 1b, S9). Considering their influence on cell migration, we here categorize the ligand presentation on nanofilaments into four levels. Self-assembled FFFIKLLI possesses super high ligand density; co-assembled FFFIKLLI and FFF at 1 to 1 and 1 to 2 ratios possesses high ligand density; co-assembly at 1 to 4 ratio possesses intermediate ligand density; and co-assembly at 1 to 249 ratio possesses low ligand density.

[Can the authors comment on any physical/chemical mechanisms that cause the restriction on the formation of protrusions? Are entropic effects/entropic penalties of any significance during this process?]

We thank the reviewer's constructive comments. In regard of the chemical mechanism, biologists have demonstrated that integrin-mediated adhesion regulated cell membrane protrusion through the Rho family of GTPase (Mol. Biol. Cell, 2001, 12, 265-277). We conducted related experiments and it was revealed that self-assembled FFFIKLLI formed excessive binding interaction with integrin $\alpha_3\beta_1$ suppressing RhoA activity (as indicated below) causing the restriction on the formation of protrusions.

Fig. S30.

Representative FRET/CFP ratio images and quantitative analysis of HuH-7 cells expressing RaichuEV-Rac1 or DORA-RhoA with or without the treatment of peptide assemblies for 12 hrs. The images were coded according to a pseudo color scale, which ranges from yellow to purple with an increase in FRET activity. Scale bars represent 20 μm . $n=50$ cells for each group. Kruskal-Wallis with Dunn's multiple comparisons test was used for analysis of the data. Error bars represent s.d..

In regard of the physical mechanism, it's hard for us to make comments on the entropic effects/entropic penalties related to the protrusion restriction process since we are not familiar to this field. We will be more than happy to cooperate with physicists who are interest in such topics. And hopefully, we will be able to address the reviewer's comments in the future.

[-As a minor side note, I believe the authors have a typo on line 132: Therefore, to "assess".... We thank the reviewer, and the typo has been corrected.

[-The Discussion section reads more like a "Conclusion" section. I find minimal discussion about

the physics behind why the authors observe specific results throughout the manuscript. In the current version, the paper reads as a summary of figures and results, without significant explanations about the mechanisms employed/discovered in this work. The Discussion section needs to be reworked to reflect the novelty of the work more significantly.]

We thank the reviewer's constructive suggestions. To address that, we revised the Conclusion section as below to reflect the novelty of this research.

Original:

In summary, we developed a bottom-up nanofabrication technique for precise control of ligand presentation achieving by far the highest spatial resolution, which is beyond the submicron limitation of classic top-down techniques, and the highest surface ligand density in biomaterials without involving complex processing. We succeeded in producing uniformed nanofilaments with high to low ligand density via simple steps, achieved biphasic control of cancer cell motility from outside-in path, which could be overcome via Rac1 activation as an inside-out exit (Figure 5j). The produced super high-density ligands established an excessive binding affinity with integrin $\alpha_1\beta_3$ inducing segmentation of FA complex preventing trailing edge retraction, which selectively suppressed cancer cell migration. To promote a wide application of the technique, we randomly selected peptide sequences derived from ECM components possessing different binding interests to integrin isoforms for extracellular constructs.(32, 33) Fibronectin-derived GRGDSP, LRGDN, and synergy peptide PHSRN, laminin β_1 chain-derived YIGSR,(34) and α_1 chain-derived IKVAV, targeting integrin $\alpha_5\beta_1$, $\alpha_v\beta_3$, $\alpha_3\beta_1$, or $\alpha_6\beta_1$, were coupled with FFF obtaining a series of assembling ligands. Upon the treatment of these assembling ligands, cells phenocopied the morphology and motility of FFFIKLLI-treated cells (Figure S24) indicating that the design strategy could be applied as generalized tool to fabricate biomaterials with therapeutic potentials in targeting subtype malignant tumor associated with different integrin expression pattern.

Revised:

The conceptual exploration of constructing synthetic soft matters displaying highly ordered patterns with various symmetries is essential to hierarchical design of advanced materials. In this research, we attempt to survey engineering efforts in molecular assembly, using synthetic chemistry-the rich tool set to create assembling building blocks, for the construction of bioactive materials via bottom-up approach. Unlike most other self-assembled soft matters that utilize single component, we introduced co-assembly to refashion the molecular self-assembly into advanced nanofabrication technique for the construction of structurally and functionally more complex materials achieving biphasic control of cancer cell motility (Figure 5j). The association of functional building block in a precise manner pushes the boundary of nanofabrication producing super high ligand density via self-assembly surpassing the existing record inducing interesting cellular response that has never been reported. Via the exploration of fundamental biological questions, a novel therapeutic strategy against metastasis is unveiled.

To prove the concept, we randomly selected peptide sequences derived from ECM components possessing different binding interests to integrin isoforms for extracellular constructs.(32, 33) Fibronectin-derived GRGDSP, LRGDN, and synergy peptide PHSRN, laminin β_1 chain-derived YIGSR,(34) and α_1 chain-derived IKVAV, targeting integrin $\alpha_5\beta_1$, $\alpha_v\beta_3$, $\alpha_3\beta_1$, or $\alpha_6\beta_1$, were coupled with FFF obtaining a series of assembling ligands for the

construction of super high ligand densities. Upon these treatments, cells phenocopied the morphology and motility of FFFIKLLI-treated cells (Figure S35) indicating that the design strategy could be applied as generalized tool to fabricate biomaterials with therapeutic potentials in targeting subtype malignant tumor associated with different integrin expression pattern. Eventually, the advancement of molecular assembly may become a turning point of programmed matter to explore new modalities and to incorporate new findings in pharmacological biology.

Revision of the main text:

Original:

Self-assembly of the assembling ligand forms nanofilaments exhibiting super high ligand density.

Revised:

The assembling ligands self-assemble into nanofilaments displaying super high ligand density.

Original:

Via introducing the non-functional assembling motifs, co-assembled nanofilaments with precisely controlled ligand densities are produced by varying the proportion of the two components.

Revised:

By mixing non-functional assembling motif with assembling ligand at different proportions, the co-assembled nanofilaments displaying precisely controlled ligand densities are produced.

REVIEWERS' COMMENTS

Reviewer #1 (Remarks to the Author):

I feel that the manuscript has improved significantly. It is appropriate for publication.

Reviewer #2 (Remarks to the Author):

Following the first round of reviews, the authors have made notable adjustments to the manuscript to clarify the results of the study and expand on their work's significance. I only have two additional comments to be addressed:

-With regards to the MD models, in the new Figure S15 (c) and correspondingly in step 5 of the MD procedure, why do the authors use a square grid for the estimation of ligand distances? Can't they get a more accurate value by considering a statistical average from particle trajectories in MD simulations, as it is stated in step 6? This point is still confusing within the MD procedure and needs clarification.

-On a minor note, the revised Manuscript could benefit from adding a concise "Conclusion" section that reflects on potential future research directions, especially for understanding the underlying physical mechanisms that the authors acknowledge can be further investigated.

The following are our point-by-point responses to the comments (*in italics*) of the reviewers and the changes (underlined) in the manuscript.

REVIEWERS' COMMENTS

Reviewer #1 (Remarks to the Author):

[I feel that the manuscript has improved significantly. It is appropriate for publication.]
We thank the reviewer's positive comments.

Reviewer #2 (Remarks to the Author):

[Following the first round of reviews, the authors have made notable adjustments to the manuscript to clarify the results of the study and expand on their work's significance.]

We thank the reviewer's positive feedback. Following the comments, we addressed the reviewer's questions point-to-point as below.

[I only have two additional comment to be addressed:

-With regards to the MD models, in the new Figure S15 (c) and correspondingly in step 5 of the MD procedure, why do the authors use a square grid for the estimation of ligand distances? Can't they get a more accurate value by considering a statistical average from particle trajectories in MD simulations, as it is stated in step 6? This point is still confusing within the MD procedure and needs clarification.]

We thank the reviewer's comment. The reason why we didn't go through the same simulation process for peptide assembly with low ligand density, including FFFIKLLI:FFF = 1:179, 1:89, and 1:44, exactly as for high ligand density but a square grid is because they are out of the focus of this article, we only did a rough estimation based on FFF unit cell conformation.

In this research, we intended to examine whether peptide assembly technique could be utilized in the fabrication of nanostructures with super high integrin ligand density, especially the ones beyond the density that has been achieved by other nanofabrication techniques. Therefore, we did the MD simulation of self-assembled FFFIKLLI and co-assembled FFFIKLLI:FFF at 1:1 to 1:4 ratios since they had high ligand densities. The simulation results are consistent to the experimental results indicating that self-assembled FFFIKLLI creates super high ligand density beyond the technical limit of top-down nanofabrication, inducing a unique cell motility that has never been well reported. While upon the treatment of FFFIKLLI:FFF at 1:179, 1:89, and 1:44 ratios, the integrin activities are the phenotypes that have been well studied and reported. Besides that, none of them is the critical ratio inducing transitions between the integrin activity phenotypes. Therefore, in regard of the complexity of simulating bigger groups of molecules and the limit capacity of our computers, we only did a rough estimation based on the unit cell conformation of FFF. The purpose of the estimation is to examine whether the ligand density is decreased by raising the proportion of FFF. Together with the experimental results, the rough estimations are consistent to the reported studies that enhanced cell motility could be achieved by reducing ligand density.

Although we did not conduct a series of sophisticated MD simulations of co-assemblies with low ligand densities in this research article, we believe that MD simulations of the whole set of co-assemblies correlating with the integrin activities and the transition between phenotypes will provide mechanistic insights in regard of the control of integrin activities via peptide assemblies. Therefore, we are characterizing a series of engineered integrin ligands as reported in the research article (Supplementary Figure 35). Together with the MD simulation results of their co-assemblies with FFF at various ratios, we intend to provide more insights in regard of the control of ligand density and ligand distance via peptide assembly in our coming articles.

To address the reviewer's comments here, we revised the related content in the Supplementary Information as indicated below to clarify the simulation process for co-assemblies with low ligand densities:

Original:

5. After extending the structure along the unit axes, the surface that exposes most integrin ligand IKLLI was presented in a defined square area ($10 \times 10 \text{ nm}^2$) within the dimension range of nanofilaments (Fig. S11). The ligand density on the surface of nanofilaments formed by co-assembly of FFFIKLLI and FFF at 1:44, 1: 89, and 1:249 ratio was estimated statistically based on the crystal structure of FFF unit cell by replacing one FFF with FFFIKLLI on the filament that composed of 45, 90, and 250 FFF, respectively. The detailed calculation results are presented in Fig. S12.

Revised:

5. After extending the structure along the unit axes, the surface that exposes most integrin ligand IKLLI was presented in a defined square area ($10 \times 10 \text{ nm}^2$) within the dimension range of nanofilaments (Supplementary Figure 14). The ligand density on the surface of nanofilaments formed by co-assembly of FFFIKLLI and FFF at 1:44, 1: 89, and 1:249 ratio was roughly estimated statistically based on the crystal structure of FFF unit cell by replacing one FFF with FFFIKLLI on the filament that composed of 45, 90, and 250 FFF, respectively. The alteration of packing dimension induced by the insertion of FFFIKLLI was ignored since the majority of the packing molecules are FFF. Because the C-terminus of FFF that can be covalently linked with IKLLI only exposes toward the zx plain, we took the area size of this plain for surface estimation. The detailed calculation results are presented in Supplementary Figure 15.

Original:

Fig. S15. (a) Space-filling model of crystal structure unit cell of FFF at zy, zx, xy plain. FFF motif was presented in pink. (b) Because the C-terminus of FFF that can be covalently linked with IKLLI is only exposed toward the zx plain (the image in the middle), we took the area size of this plain for surface calculation. The calculation results of ligand density for FFFIKLLI/FFF at 1 to 249 to 1 to 44 were summarized. (c) The scheme represents the estimation of ligand distance in regard of three different densities.

Revised:

Supplementary Figure 15. Estimation of ligand density in various peptide assemblies.

(a) Space-filling model of crystal structure unit cell of FFF at zy, zx, xy plain. FFF motif was presented in pink. (b) The rough estimation of ligand density was carried out based on the conformation of the unit cell of FFF. The alteration of packing parameters induced by the insertion of FFFIKLLI was ignored since the majority of the packing molecules are FFF. Because the C-terminus of FFF that can be covalently linked with IKLLI is only exposed toward the zx plain (the image in the middle), we took the area size of this plain for surface estimation. The calculation results of ligand density for FFFIKLLI/FFF at 1 to 249 to 1 to 44 were summarized. (c) The scheme represents the rough estimation of ligand distance in regard of three different densities.

[-On a minor note, the revised Manuscript could benefit from adding a concise "Conclusion" section that reflects on potential future research directions, especially for understanding the underlying physical mechanisms that the authors acknowledge can be further investigated.]

We thank the reviewer's constructive suggestion. Following that, we revised the Conclusion section as below:

Original:

The conceptual exploration of constructing synthetic soft matters displaying highly ordered patterns with various symmetries is essential to hierarchical design of advanced materials. In this research, we attempt to survey engineering efforts in molecular assembly, using synthetic chemistry-the rich tool set to create assembling building blocks, for the construction of bioactive materials via bottom-up approach. Unlike most other self-assembled soft matters that utilize single component, we introduced co-assembly to refashion the molecular self-assembly into advanced nanofabrication technique for the construction of structurally and functionally more complex materials achieving biphasic control of cancer cell motility (Figure 5j). The association of functional building block in a precise manner pushes the boundary of nanofabrication producing super high ligand density via self-assembly surpassing the existing record inducing interesting cellular response that has never been reported. Via the exploration of fundamental biological questions, a novel therapeutic strategy against metastasis is unveiled.

To prove the concept, we randomly selected peptide sequences derived from ECM components possessing different binding interests to integrin isoforms for extracellular constructs.(32, 33) Fibronectin-derived GRGDSP, LRGDN, and synergy peptide PHSRN, laminin β 1 chain-derived YIGSR,(34) and α 1 chain-derived IKVAV, targeting integrin $\alpha_5\beta_1$, $\alpha_v\beta_3$, $\alpha_3\beta_1$, or $\alpha_6\beta_1$, were coupled with FFF obtaining a series of assembling ligands for the construction of super high ligand densities. Upon these treatments, cells phenocopied the morphology and motility of FFFIKLLI-treated cells (Figure S35) indicating that the design strategy could be applied as generalized tool to fabricate biomaterials with therapeutic potentials in targeting subtype malignant tumor associated with different integrin expression pattern. Eventually, the advancement of molecular assembly may become a turning point of programmed matter to explore new modalities and to incorporate new findings in pharmacological biology.

Revised:

The conceptual exploration of constructing synthetic soft matters displaying highly ordered patterns with various symmetries is essential to hierarchical design of advanced materials. In this research, we attempt to survey engineering efforts in molecular assembly, using synthetic chemistry-the rich tool set to create assembling building blocks, for the construction of bioactive materials via bottom-up approach. Unlike most other self-assembled soft matters that utilize single component, we introduced co-assembly to refashion the molecular self-assembly into advanced nanofabrication technique for the construction of structurally and functionally more complex materials achieving biphasic control of cancer cell motility (Figure 5j). The association of functional building block in a precise manner pushes the boundary of nanofabrication producing super high ligand density via self-assembly surpassing the existing record inducing interesting cellular response that has never been reported. Via the exploration of fundamental biological questions, a novel therapeutic strategy against metastasis is unveiled.

To prove the concept, we randomly selected peptide sequences derived from ECM components possessing different binding interests to integrin isoforms for extracellular constructs.(32, 33) Fibronectin-derived GRGDSP, LRGDN, and synergy peptide PHSRN, laminin β 1 chain-derived YIGSR,(34) and α 1 chain-derived IKVAV, targeting integrin $\alpha_3\beta_1$, $\alpha_v\beta_3$, $\alpha_3\beta_1$, or $\alpha_6\beta_1$, were coupled with FFF obtaining a series of assembling ligands for the construction of super high ligand densities. Upon these treatments, cells phenocopied the morphology and motility of FFFIKLLI-treated cells (Supplementary Figure 35) indicating that the design strategy could be applied as generalized tool to fabricate biomaterials with therapeutic potentials in targeting subtype malignant tumor associated with different integrin expression pattern. Following the experimental observations, the insertion of mechanistic insights of dynamic interactions between the peptide assemblies and the integrin clustering from molecular dynamic simulations will guide the optimization of ligand engineering. Eventually, the advancement of molecular assembly may become a turning point of programmed matter to explore new modalities and to incorporate new findings in pharmacological biology.